# Ozone depletion events in the Arctic spring of 2019: A new modeling approach to bromine emissions

Maximilian Herrmann[1], Moritz Schöne[2,3], Christian Borger[2], Simon Warnach[2,3], Thomas Wagner[2,3], Ulrich Platt[3,4], and Eva Gutheil[1,4]

[1]Interdisciplinary Center for Scientific Computing, Heidelberg University, Heidelberg University, Im Neuenheimer Feld 205, Heidelberg 69120, Germany
[2]Max-Planck Institute for Chemistry, Mainz, Hahn-Meitner-Weg 1, Mainz 55128, Germany
[3]Institute of Environmental Physics, Heidelberg University, Im Neuenheimer Feld 229, Heidelberg 69120, Germany
[4]Heidelberg Center for the Environment, Heidelberg University, Im Neuenheimer Feld 130.1, Heidelberg 69120, Germany

**Correspondence:** Eva Gutheil (gutheil@iwr.uni-heidelberg.de)

**Abstract.** Ozone depletion events (ODEs) are a common occurence in the boundary layer during Arctic spring. Ozone is depleted by bromine species, which are most likely emitted from snow, sea ice or aerosols in an auto-catalytic reaction cycle. Previous three-dimensional modeling studies of ODEs assumed an infinite bromine source at the ground. In the present study, an alternative emission scheme is presented in which a finite amount of bromide in the snow is tracked over time. For this purpose, a modified version of the **W**eather **R**esearch and **F**orecasting model coupled with **Chem**istry (WRF-Chem) is used to study ODEs in the Arctic from February to May 2019. The model data are compared to in-situ measurements, ozone sonde flights as well as satellite data. A simulation of the ODEs in the Arctic spring of 2009 using the infinite bromide assumption on FY ice is transferred to the spring of 2019 which achieves good agreement with the observations, however, there is some disagreement in April 2009 and 2019 with respect to an overestimation concerning both the magnitude and the number of ODEs. New simulations using the finite bromide assumption greatly improve agreement with in-situ observations at Utqiaġvik, Alaska, Zeppelin Mountain, Svalbard, and Pallas, Finland in April 2019, suggesting that bromide on the sea ice is depleted to an extent that reduces the bromine release. The new simulations also slightly improve the agreement with observations at these sites in February and March. A comparison to measurements near Eureka, Canada and Nord Station, Greenland shows that multi-year ice and possibly snow-covered land may be significant bromine sources. However, assuming higher releasable bromide near Eureka does not remove all disagreement with the observations. The numerical results are also compared to tropospheric BrO vertical column densities generated with a new retrieval method from TROPOMI observations. BrO VCDs above $5 \times 10^{13}$ molec/cm$^2$ observed by the satellite agree well with the model results. However, the model also predicts BrO VCDs of around $3 \times 10^{13}$ molec/cm$^2$ throughout the Arctic and patches of BrO VCDs of around $10^{14}$ molec/cm$^2$ not observed by the satellite, especially near Hudson Bay. This suggests that snow at Hudson Bay may be a weaker bromine source in late spring compared to snow in the north.

# 1 Introduction

Ozone depletion events (ODEs) commonly occur in the Arctic boundary layer during spring. The ozone mixing ratio is reduced from its background level of approximately 30-60 nmol/mol to possibly zero levels coinciding with an increase in the bromine concentrations (Oltmans, 1981; Bottenheim et al., 1986; Barrie et al., 1988; Hausmann and Platt, 1994; Wagner and Platt, 1998; Richter et al., 1998; Wagner et al., 2001; Frieß et al., 2004; Wagner et al., 2007; Helmig et al., 2012; Halfacre et al., 2014; Zhao et al., 2016; Blechschmidt et al., 2016; Seo et al., 2019, 2020; Bougoudis et al., 2020). The ozone depletion is of special interest since ozone is a very important trace gas due to its role in air pollution and its high oxidation potential. Additionally, the bromine released during an ODE can oxidize mercury (Dastoor et al., 2008; Steffen et al., 2008), which may further pollute the Arctic ecosystem. The most important ozone depletion cycle is catalyzed by bromine (Barrie et al., 1988; Wang et al., 2019)

$$Br + O_3 \longrightarrow BrO + O_2 \tag{R1}$$

$$BrO + BrO \longrightarrow 2\,Br + O_2 \tag{R2}$$

The rate limiting step is typically the BrO self-reaction (R2), which means that the reaction rate of the net reaction

$$2\,O_3 \longrightarrow 3\,O_2 \tag{R3}$$

is quadratic in the concentration of BrO. A bromine atom can be recycled approximately 100 times in the reaction cycle (R1–R3), yielding a high potential for ozone destruction before it is converted to the chemically inert species HBr, see Reaction (R17). The recycling of BrO and thus ozone depletion can also occur under sunlight in cross-reactions with other halogens, primarily ClO and IO, with approximately one order of magnitude higher reaction rates (Atkinson et al., 2007)

$$BrO + XO \longrightarrow BrX + O_2 \tag{R4}$$

$$BrX + h\nu \longrightarrow Br + X \tag{R5}$$

Again with sunlight, an alternative recycling path of BrO involving $HO_2$ is possible

$$BrO + HO_2 \longrightarrow HOBr + O_2 \tag{R6}$$

$$HOBr + h\nu \longrightarrow Br + OH \tag{R7}$$

Since OH can be converted to $HO_2$ e.g., by reaction with CO, reactions (R6) and (R7) constitute an ozone destruction mechanism the rate of which varies linear with BrO.

Bromide ($Br^-$) stored as sea salt in reservoirs such as snow, sea ice or aerosol particles is the most likely source of the gaseous bromine (Fan and Jacob, 1992; McConnell et al., 1992; Platt and Janssen, 1995; Pratt et al., 2013; Simpson et al., 2015; Custard et al., 2017). Several activation mechanisms of the bromide have been proposed. The most widely accepted, heterogeneous and auto-catalytic 'bromine explosion' mechanism (Platt and Janssen, 1995; Platt and Lehrer, 1997; Wennberg, 1999) is capable of a quick activation of bromide. It consists of the Reactions (R1), (R6), the heterogeneous reaction

$$HOBr\,(g) + H^+\,(aq) + Br^-\,(aq) \longrightarrow Br_2\,(g) + H_2O\,(l) \tag{R8}$$

and a photolysis reaction

$$Br_2 + h\nu \longrightarrow 2\,Br\cdot \tag{R9}$$

From the net reaction

$$Br\,(g) + O_3\,(g) + HO_2\,(g) + Br^-\,(aq) + H^+\,(aq) + h\nu \longrightarrow 2\,Br\,(g) + 2\,O_2\,(g) + H_2O\,(l), \tag{R10}$$

it can be seen that the quantity of bromine atoms in the gas-phase will double after each reaction cycle, leading to an exponential increase of the BrO concentration. Another mechanism similar to the bromine explosion involves nitrogen oxide, where Reactions (R6) and (R8) are replaced by

$$BrO + NO_2 + M \longrightarrow BrONO_2 + M \tag{R11}$$

$$BrONO_2\,(g) + Br^-\,(aq) \longrightarrow Br_2\,(g) + NO_3{}^-\,(aq) \tag{R12}$$

Three requirements for the bromine explosion can be seen from the net reaction (R10): Since $H^+$ ions are consumed in the reaction cycle, a pH-dependence of the bromine explosion cycle is evident. Fickert et al. (1999) suggested that a pH-value of below 6.5 is required for fast reactions in the liquid phase, which is supported by field-based and lab-based experiments (Pratt et al., 2013; Wren et al., 2013; Halfacre et al., 2019). Additionally, due to the photolysis reaction of $Br_2$, the bromine explosion cannot occur without sunlight. Moreover, the bromine explosion cannot occur without reactive bromine already present in the gas-phase, so that an additional trigger producing the first bromine is essential.

Additional pathways for releasing bromine, which may also serve as an initial trigger of the bromine explosion, are detailed in the following. Initial bromine may be activated by $N_2O_5$ (Bertram and Thornton, 2009; Lopez-Hilfiker et al., 2012)

$$N_2O_5\,(g) + Br^-\,(aq) \longrightarrow BrNO_2\,(g) + NO_3{}^-\,(aq) \tag{R13}$$

Under sunlight, $BrNO_2$ is photolyzed

$$BrNO_2 + h\nu \longrightarrow Br + NO_2\cdot \tag{R14}$$

The bromide oxidation by ozone (Oum et al., 1998; Artiglia et al., 2017), which is likely to only occur efficiently under sunlight (Pratt et al., 2013), may also trigger bromine explosions

$$O_3\,(g) + H^+\,(aq) + 2\,Br^-\,(aq) \longrightarrow Br_2\,(g) + H_2O\,(aq) + O_2\,(g) \tag{R15}$$

Bromine may additionally be released by a reaction of OH with bromide inside the snow under sunlight (Sjostedt and Abbatt, 2008; Pratt et al., 2013; Halfacre et al., 2019)

$$OH\,(aq) + 2\,Br^-\,(aq) + H^+\,(aq) + h\nu \longrightarrow Br_2\,(g) + H_2O\,(l) \tag{R16}$$

Blowing snow, snow particles lifted into the air by strong winds, might sublimate and produce fresh salt aerosols, which may serve as a large surface for bromine activation (Yang et al., 2008, 2010; Blechschmidt et al., 2016; Zhao et al., 2016; Huang and

Jaeglé, 2017; Yang et al., 2019). Frost flowers were previously discussed as a major bromide source (Kaleschke et al., 2004; Alvarez-Aviles et al., 2008), but later studies found this to be unlikely (Obbard et al., 2009; Pratt et al., 2013).

Since Reaction (R1) slows down for a smaller ozone mixing ratio, the reactions of atomic bromide with species such as aldehydes, producing chemically inert HBr, become major pathways. One example is the reaction of Br with formaldehyde, HCHO

$$Br + HCHO + O_2 \longrightarrow HBr + CO + HO_2 \cdot \tag{R17}$$

HBr can then be brought back to the bromide reservoir through depositions to the surfaces of aerosols and snow.

Several meteorological factors influence the occurrence of ODEs. Many of the reaction cycles mentioned above involve photolysis reactions requiring sunlight, which is why ODEs are not observed during winter.

Lower temperatures have been found to increase the frequency of ODEs in some studies (Tarasick and Bottenheim, 2002; Pöhler et al., 2010; Seo et al., 2020). However, ODEs were also observed at higher temperatures (Bottenheim et al., 2009) and without an apparent temperature dependence (Halfacre et al., 2014), with the exception that ODEs do not occur above 0°C when snow melts (Burd et al., 2017). ODEs were found to occur more commonly in stable boundary layers (Wagner et al., 2001; Frieß et al., 2004; Lehrer et al., 2004; Koo et al., 2012; Zhao et al., 2016). There is very little vertical mixing under stable conditions, so that surface emissions are transported away from the surface and diluted less quickly in comparison to other boundary layer configurations. As a result, gas-phase bromine concentrations in the boundary layer increase, accelerating both further bromine emissions and the ozone depletion.

In outdoor snow chamber experiments, Pratt et al. (2013) found photo-chemical production of reactive bromine from Arctic surface snow, but not from sea ice. Additionally, the type of surface covered by snow may be an important factor. Snow covering first-year (FY) ice, ice freshly formed in the previous winter, was found to correlate with bromine producing sites (Simpson et al., 2007; Abbatt et al., 2012; Bougoudis et al., 2020) in contrast to multi-year (MY) ice. Despite that, bromine activation over MY ice was observed as well (Peterson et al., 2019). Finally, large bromine concentrations in the snow and bromine emissions from the snow-covered tundra were observed (Simpson et al., 2005; Pratt et al., 2013; Custard et al., 2017; McNamara et al., 2020). Peterson et al. (2018) measured near-surface BrO up to 200 km inland during the **BR**omine, **O**zone, and **M**ercury **EX**periment (BROMEX) flights, which suggests that even snow far away from the coast may be an active source of bromine.

Due to the coupling of both meteorology and chemistry for ozone depletion and bromine explosion events, only three-dimensional models can simulate all of the relevant processes. Recent studies investigated two different major pathways of bromine emissions: On the one hand several studies focused on bromide activation on aerosols formed from sublimation of snow particles lifted into the air by blowing snow events (Yang et al., 2008, 2010; Huang and Jaeglé, 2017; Yang et al., 2019, 2020). They found that the inclusion of the blowing snow emissions greatly improved Arctic surface ozone seasonality reproduction. Other studies instead focused on emissions of bromine due to heterogeneous reactions on surface snow, which was first developed by Lehrer et al. (2004) and adapted to three-dimensional models by Toyota et al. (2011). The mechanism was implemented into the 3D air quality model **G**lobal **E**nvironmental **M**ultiscale model with **A**ir **Q**uality processes (GEM-AQ) by Toyota et al. (2011) and subsequently into the **ECHAM/MESS**y **A**tmospheric **C**hemistry (EMAC) model

by Falk and Sinnhuber (2018) as well as the Community Atmosphere Model with Chemistry by (Fernandez et al., 2019).
The studies showed good agreement to observations at various sites and to BrO VCDs observed with the GOME satellite. WRF-Chem (Grell et al., 2005; Skamarock et al., 2008) was modified in two recent studies to allow the modeling of tropospheric ODEs. Herrmann et al. (2021) modeled ODEs in the spring of the year 2009 with bromine emissions on surface snow following Toyota et al. (2011) and found good agreement to tropospheric vertical column densities (VCD) of BrO observed by the GOME-2 satellite and other measurements of BrO and ozone. With an enhanced emission rate, the agreement with observations improved. Furthermore, the bromide oxidation by ozone was found to emit little bromine in comparison to the bromine explosion, but served as its primary trigger. Marelle et al. (2021) also modified WRF-Chem to allow the modeling of ODEs and first implemented both the surface snow emission mechanism (Toyota et al., 2011) and the blowing snow emissions (Yang et al., 2008) into a single model. Marelle et al. (2021) simulated the Arctic spring of 2012 and found blowing snow to be a strong source of sea salt aerosols. The use of the blowing snow emission mechanism was not suitable to explain the majority of the ozone depletion events whereas surface emissions were found to be the major driver of ozone depletion events in most of the Arctic. These three-dimensional models did not include the bromide content of the snow but used the assumption of infinite bromide stored in the snow pack.

Snow models with finite bromide concentrations were implemented into one-dimensional models (Thomas et al., 2011; Toyota et al., 2014), albeit at significant computational cost due to the modeling of the snow by additional grid cells with their individual chemistry mechanism and transport models, as each additional snow layer might be of similar cost as adding an additional vertical layer above ground and the implementation of a snow model into WRF-Chem is not straightforward.

In the present study, a simulation of the ODEs in the Arctic spring of 2009 (Herrmann et al., 2021) using the infinite bromide assumption on FY ice is transferred to the spring of 2019 which achieves good agreement with the observations, however, there is some disagreement in April 2009 and 2019 with respect to an overestimation concerning both the magnitude and the number of ODEs. Therefore, the surface emission mechanism used by Herrmann et al. (2021) is extended by an alternative set of assumptions, allowing for a finite bromide surface concentration in the snow to be tracked over time, which relaxes the previous assumption of infinite bromide on FY ice and zero bromide on other surfaces. The results are compared to a new tropospheric BrO vertical column density product retrieved from measurements of the **TROPO**spheric **M**onitoring **I**nstrument (TROPOMI) as well as in-situ measurements and ozone sonde flights at a number of Arctic sites.

## 2  Model description

In this section, the model used in the present study, including the new emission mechanism, is described.

### 2.1  Model setup

The compressible, non-hydrostatic and moist Euler equations on a rotating sphere together with chemistry are solved using the regional, three-dimensional and time-dependent weather prediction system **W**eather **R**esearch and **F**orecasting model coupled with **Chem**istry (WRF-Chem) 3.9.0. The MOZART mechanism (Emmons et al., 2010) extended with bromine chem-

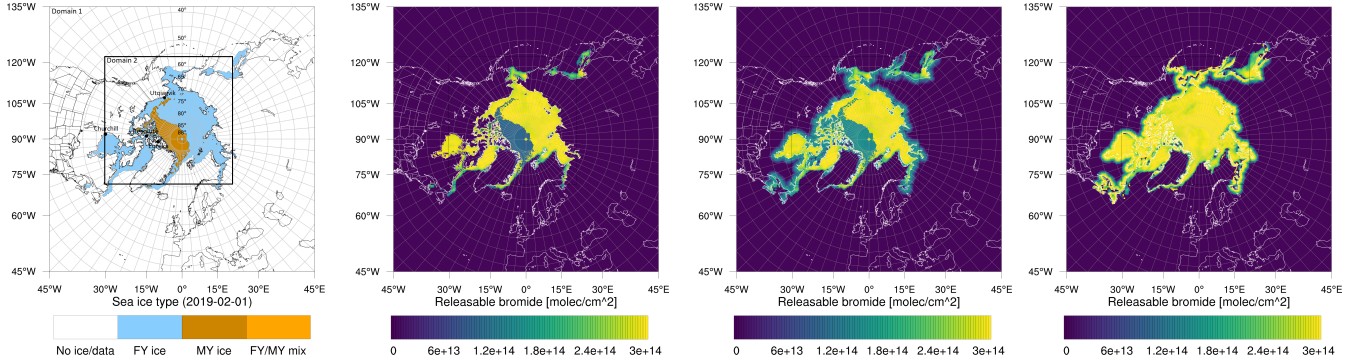

**Figure 1.** From left to right: Sea ice type data (Aaboe et al., 2017) and initial releasable bromide for the low bromide (left center), medium bromide (right center) and high bromide (right) simulations

istry (Herrmann et al., 2019) for a total of 103 gas-phase species and 359 reactions combined with four-bin sectional MOSAIC aerosols (Zaveri et al., 2008) is employed. The microphysical scheme WSM6 is employed, which is not hooked to the washout of aerosols in MOZART-MOSAIC. Wet removal of halogen species is currently not considered.

A detailed model description is given by Herrmann et al. (2021), in which the ozone depletion in the year 2009 was simulated with a slightly different setup. The following details are different to the setup of Herrmann et al. (2021). Two domains are used with the nesting method for a local grid refinement, see Fig. 1. A coarse and a fine resolution simulation, simply referred to as domain 1 and 2 hereafter, are conducted at the same time. The output of domain 1 is used online and without feedback as boundary conditions for domain 2. The ERA5 Reanalysis (Hersbach et al., 2020) for meteorology and global CAM-Chem output (Buchholz et al., 2021) for chemistry are used as boundary conditions for domain 1 and as initial conditions for both domains. Both domains are nudged to the temperature, horizontal wind speed, humidity, and surface fields of ERA5 on a one-hour timescale. Nudging is employed for the whole simulation period and is turned off inside the boundary layer. Domain 1 employs a larger timestep of 3 min compared to a 1 min timestep for domain 2, which is chosen to fulfill the Courant criterium. Domain 1 is centered on the north pole using the polar stereographic projection. Domain 1 covers a $12{,}600 \times 12{,}600$ km$^2$ area with a 60 km resolution, domain 2 a $6{,}000 \times 6{,}000$ km$^2$ area with a 20 km resolution. The 2014 Community Emissions Data System (Hoesly et al., 2018) is used as anthropogenic emissions.

## 2.2 Rescaling of the initial ozone mixing ratio

An initial comparison of the ozone mixing ratio with in-situ measurements at Utqiaġvik and Eureka revealed the initial ozone mixing ratio provided by CAM-Chem to be estimated too high by 16 ppb and 6 ppb, respectively. In order to improve the agreement of the initial values of the model to the observations, the initial ozone mixing ratio $[\text{O}_3]_{\text{CAM-Chem}}$ was rescaled in the troposphere with a dependence on latitude $\theta$, using the observations at Eureka and Utqiaġvik at ground level. Observations at other locations, including Zeppelin, Pallas, and Summit were not used. The rescaled initial ozone value at these locations is smaller than the observed ozone mixing ratio. For latitudes larger than $79.983°$, where Eureka is located, the scaling factor of

0.846 for Eureka is used. For latitudes between 71.323° and 79.983°, the latitudes of the stations at Utqiaġvik and Eureka, the factors are linearly interpolated between the factors at Utqiaġvik and Eureka. Investigative simulations suggest that the best

match is achieved when the factor at Utqiaġvik, 63.2%, is assumed from the latitude of Utqiaġvik, 71.323°, to a latitude of around 60°. For the same reason, linear scaling is again applied between 60° with the scaling factor of 63.2% to 45° with a scaling factor of unity.

$$[O_3]_{\text{model}} = [O_3]_{\text{CAM-Chem}} \times \begin{cases} 0.846 & \text{for } \theta \geq 79.983° \\ 0.632 + 0.214\,\dfrac{\theta - 71.323°}{8.66°} & \text{for } 71.323° < \theta < 79.983° \\ 0.632 & \text{for } 60° < \theta \leq 71.323° \\ 1 - 0.368\,\dfrac{\theta - 45°}{15°} & \text{for } 45° < \theta \leq 60° \\ 1 & \text{for } \theta \leq 45° \end{cases} \quad (1)$$

This rescaling of the initial ozone mixing ratio has a large impact in February, but becomes less important in March and April,

once ODEs start occurring frequently. The impact of the rescaled initial conditions fade over time, however, the impact of the first ODEs on the ozone concentration are so large, that the effect of the fading influence of the initial conditions are overshadowed. The rescaling is inactive for latitudes lower than 45°, so that the boundary conditions of domain d01 are not affected by the rescaling.

## 2.3   Heterogeneous bromine emission mechanism

The following bromine emitting heterogeneous reactions on snow surfaces are implemented

$$\text{HOBr} \xrightarrow{a\,\text{Br}^- + b\,\text{Cl}^- + \text{H}^+} a\,\text{Br}_2 + b\,\text{BrCl} + \text{H}_2\text{O} \tag{R18}$$

$$\text{BrONO}_2 \xrightarrow{\text{H}_2\text{O}} \text{HOBr} + \text{HNO}_3 \tag{R19}$$

$$\text{N}_2\text{O}_5 \xrightarrow{a\,\text{Br}^- + \text{H}^+} a\,\text{BrNO}_2 + (1 + \text{b})\,\text{HNO}_3 \tag{R20}$$

$$\text{O}_3 \xrightarrow{\gamma(2\,\text{Br}^- + 2\,\text{H}^+)} a\gamma(\text{Br}_2 + \text{H}_2\text{O} + \text{O}_2) \tag{R21}$$

The emission probability of $\text{Br}_2$ due to ozone, $\gamma$, is assumed to be 7.5% for a solar zenith angle of less than 85° and 0.1% otherwise, following Toyota et al. (2011). The parameters $a$ and $b$ are the emission probabilities of $\text{Br}_2$ and BrCl, respectively. Species above the reaction arrows are part of the liquid phase on the surface, which is not directly modeled. Reactions occurring in the liquid phase are assumed to be fast in comparison to the depositions.

    The emissions are implemented as lower boundary conditions, here as an example the $\text{Br}_2$ emission flux resulting from the

deposition of HOBr

$$F_{\text{d}}(\text{Br}_2|\text{HOBr}) = a\rho_{d,0}v_{\text{d}}(\text{HOBr})\,[\text{HOBr}]_0, \tag{2}$$

$\rho_{d,0}$ is the density of dry air in the lowest grid cell, $v_{\text{d}}(\text{HOBr})$ the deposition velocity is calculated using the Wesely dry deposition module (Wesely, 1989) of WRF-Chem, with the surface resistance taken from Herrmann et al. (2019) and $[\text{HOBr}]_0$

is the HOBr mixing ratio in the lowest grid cell. The deposition scheme is only modified for the bromine species HOBr, HBr, and $BrONO_2$ species by overwriting the surface resistance following Huff and Abbatt (2000, 2002) on snow and ice surfaces, resulting in deposition velocities of 1-2 cm/s. Without these changes, WRF-Chem would calculate deposition velocities of around 0.1-0.2 cm/s for halogen species on snow or ice surfaces. Deposition velocities of halogens on non-snow and non-ice surfaces use the standard implementation of WRF-Chem.

The parameters $a$ and $b$ are calculated under two sets of assumptions. In both cases, chloride content of snow surfaces is assumed to be infinite. In the first set of assumptions, referred to as "infinite first-year ice bromide" from now on, bromide supply on FY ice is assumed to be infinite and zero on other surfaces. With these assumptions, $b = 1 - a$ on all snow covered surfaces and $a = 1$ on snow covering FY ice. On MY ice and snow covered land, $a$ depends on the depositions of HBr and the depositions of bromine emitting species

$$a = \min\left(1, \frac{F_d(\text{HBr})}{F_d(\text{HOBr}) + F_d(\text{N}_2\text{O}_5) + 2\gamma F_d(\text{O}_3)}\right) \tag{3}$$

The ozone depositions consume two bromide ions in the reservoir and emit them as $Br_2$ into the gas-phase, which explains the factor of two in front of $F_d(O_3)$ in addition to the emission probability $\gamma$. In other words, $Br_2$ emissions are limited by the HBr depositions, allowing recycling of HBr on MY and snow covered land, but not emissions of new bromine. With the second set of assumptions, the bromide content is assumed to be finite and is tracked by the surface variable $Br_{surf}$, the column density of bromide releasable from the snow, over the course of the simulation. Three different numerical simulations are conducted using the parameters $c_1, c_2, c_3$, which are concentrations of releasable bromide in snow with units $\text{molec/cm}^2$ and should be seen as free parameters, with

$$[\text{Br}_{\text{surf}}] = \begin{cases} c_1 & \text{on FY ice and } h < 1 \text{ km} \\ c_2 & \text{on MY ice and } h < 1 \text{ km} \\ c_3 & \text{on snow covered land and } h < 1 \text{ km} \\ 0 & \text{otherwise} \end{cases} \tag{4}$$

Different sets of parameters are used for three finite bromide simulations with low, medium, and high initial bromide concentrations listed in Tab. 1 and displayed in Fig. 1. In the remainder of the paper, the corresponding simulations are called 'low', 'medium', and 'high' bromide simulations.

For locations with an elevation $h$ higher than 1 km, $Br_{surf}$ is set to zero. On sea ice, $Br_{surf}$ is multiplied by the sea ice coverage ratio provided by ERA5.

Pratt et al. (2013) measured a bromide concentration of 11.4 μM for the first cm of snow covering sea ice. Thus, the top 1 cm of snow contains a bromide column density $Br_{snow}$ of approximately

$$\text{Br}_{\text{snow}} \approx 10\,\mu\text{M} \times 1\,\text{cm} \approx 10 \times 10^{-6} \times 6 \times 10^{23} \frac{\text{molec cm}}{10^3 \text{cm}^3} = 6 \times 10^{15} \frac{\text{molec}}{\text{cm}^2}. \tag{5}$$

Choosing this value as $c_1$ would lead to large BrO VCDs similar to the infinite FY ice bromide simulation. Instead, the value of $c_1 = 3 \times 10^{14}\,\text{molec/cm}^2$ was chosen over FY ice for all finite bromide simulations, since observed BrO vertical column

densities (VCD) have maximum values of approximately $10^{14}\,\mathrm{molec/cm^2}$ and typically, about one third of gaseous bromine is BrO in the present model. Using the above calculation, Krnavek et al. (2012) measured bromide column densities on FY ice of $1.2 \times 10^{14}\,\mathrm{molec/cm^2}$ to $1.8 \times 10^{17}\,\mathrm{molec/cm^2}$, with a median of $1.2 \times 10^{15}\,\mathrm{molec/cm^2}$ on thick FY ice, so that the values on FY ice used in this work are consistent with the lower range of values found by Krnavek et al. (2012). Peterson et al. (2019) found lower halide concentrations due to measuring thicker sea ice (> 1 m), and they found an average bromide column density of $3.6 \times 10^{14}\,\mathrm{molec/cm^2}$, which is consistent with the value used in this work. The values of $\mathrm{Br_{surf}}$ over MY ice and snow covered land are varied in two additional simulations. On MY ice, Krnavek et al. (2012) observed values in the range of $3.6 \times 10^{13}\,\mathrm{molec/cm^2}$ to $2.4 \times 10^{14}\,\mathrm{molec/cm^2}$ and with a median of $1.5 \times 10^{14}\,\mathrm{molec/cm^2}$, whereas Peterson et al. (2019) found average values of $2 \times 10^{14}\,\mathrm{molec/cm^2}$ and thus, the initial bromide column densities on MY ice chosen in the present work are consistent with their findings. It should be noted, that the variability of the bromide content of snow is currently unclear.

For tundra surface snow, concentrations of 0.08 to 0.4 μM and 0.04 to 0.56 μM were measured by Pratt et al. (2013) and by Krnavek et al. (2012), respectively, so the first centimeter of tundra surface snow contains a bromide column density of 2.5 to $24 \times 10^{13}\,\frac{\mathrm{molec}}{\mathrm{cm^2}}$, which roughly corresponds to the values of releasable bromide assumed for snow covered land (5 to $30 \times 10^{13}\,\frac{\mathrm{molec}}{\mathrm{cm^2}}$). For snow covering sea ice, however, assuming the values measured by Pratt et al. (2013) and median values of Krnavek et al. (2012), $\mathrm{Br_{snow}}$ is approximately 20 times larger than the releasable bromide $\mathrm{Br_{surf}}$ used in this study. A release of 5% of the total bromide only slightly changes the bromide to chloride ratio, but might significantly increase the pH and inhibit further release. For the low bromide simulation, snow covered land is determined by the USGS 30s landuse category 'Snow and Ice' and a sea ice coverage of zero. As can be seen in Fig. 1, only few land surfaces are considered to be snow covered. The WRF-Chem snow cover prediction is not used in this work, since it could lead to bromine emission even very far from the coast. The model predicts snow cover in most of Canada, Scandinavia and even the northern USA for a significant period of the simulation. Additionally, for snow close to the coast, it may still be assumed that the bromide is from sea salt. In two additional simulations with the finite bromide assumption, land with a distance of at most 300 km to sea ice is considered to be covered by snow. The bromide level on snow-covered land is reduced with distance to sea ice, with the full value being used up to 100 km inland and then the value is linearly decreased to zero from 100 km to 300 km inland. During BROMEX flights in 2012, Peterson et al. (2018) found enhanced BrO levels near the surface up to 200 km inland, which serves as the motivation for the value mentioned above. Releasable bromide on MY ice and snow-covered land are increased to half and the full value on FY ice, respectively. An overview of the simulations is given in Tab. 1.

At each model time $n$ with timestep $\mathrm{d}t$, HBr depositions are added to the releasable bromide

$$\mathrm{Br_{surf}}(n+1) = \mathrm{Br_{surf}}(n) + \mathrm{d}t F_\mathrm{d}(\mathrm{HBr}) \tag{6}$$

In order to calculate $a$, the bromide atoms which may be released by depositions of HOBr, ozone and $\mathrm{N_2O_5}$ in this timestep are diagnosed

$$\mathrm{Br_{rel}} = \mathrm{d}t \left( F_\mathrm{d}(\mathrm{HOBr}) + F_\mathrm{d}(\mathrm{N_2O_5}) + 2\gamma F_\mathrm{d}(\mathrm{O_3}) \right) \tag{7}$$

**Table 1.** Overview of the four simulations. Initial releasable bromide, see Eq. (1).

| Simulation name | Initial releasable bromide, $c_1, c_2, c_3$ [$10^{14}$ molec/cm$^2$] | Definition of snow-covered land | simulation start |
|---|---|---|---|
| infinite first-year ice bromide | $\infty$, 0, 0 | USGS 30s landuse | February 1, 2019 |
| Low bromide | 3, 1, 0.5 | USGS 30s landuse | February 1, 2019 |
| Medium bromide | 3, 1.5, 1.5 | 300 km distance to sea ice | February 1, 2019 |
| High bromide | 3, 3, 3 | 300 km distance to sea ice | April 1, 2019 |

If releasable bromide is sufficient, $\mathrm{Br}_{\mathrm{surf}}(n) > \mathrm{Br}_{\mathrm{rel}}$, all diagnosed bromide atoms can be released. $a$ is then set to one and the released bromide atoms are substracted from the releasable bromide

$$\mathrm{Br}_{\mathrm{surf}}(n+1) = \mathrm{Br}_{\mathrm{surf}}(n) - \mathrm{Br}_{\mathrm{rel}} \tag{8}$$

Otherwise, all releasable bromide will be released, so that $\mathrm{Br}_{\mathrm{surf}}(n+1) = 0$ and

$$a = \frac{\mathrm{Br}_{\mathrm{surf}}(n)}{\mathrm{Br}_{\mathrm{rel}}}. \tag{9}$$

Basically, $\mathrm{Br}_2$ emissions are limited by the amount of available releasable bromide. Exploratory simulations showed that a replenishment of releasable bromide over time is necessary, since otherwise most bromide would be depleted over the course of March. Possible mechanisms for the replenishment of bromide are wind-transport of sea spray, upward migration from sea ice and wind-blown frost flowers (Domine et al., 2004). For this purpose, bromide is replenished by a constant rate of its initial value divided by a timescale of one week, which on FY ice results in a replenishment rate of

$$\mathrm{Br}_{\mathrm{repl}} = \frac{3 \times 10^{14}\,\mathrm{molec/cm^2}}{604800\,\mathrm{s}} \approx 5 \times 10^8\,\mathrm{molec/(cm^2 s)} \tag{10}$$

In some situations, the replenished bromide is immediately released, which allows a comparison to emission rates. Typical reported emission rates are on the orders of $10^7$ to $10^9\,\mathrm{molec/(cm^2 s)}$ (Custard et al., 2017; Wang and Pratt, 2017). For the year 2009, Herrmann et al. (2021) found typical $\mathrm{Br}_2$ emissions of 2 to $8 \times 10^9\,\mathrm{molec/(cm^2 s)}$ during daytime in late March and April. The initial values of releasable bromide, cf. Eq. (1), the replenishment rate in Eq. (10) and the definition of snow-covered land should be seen as free model parameters.

The simulations take place over approximately 13 weeks. With the replenishment timescale of one week, the value of $3 \times 10^{14}\,\mathrm{molec/(cm^2 s)}$ bromide assumed on FY ice can be at most released 14 times (initial concentration + replenishment over 13 weeks). The upper limit of bromine that may be released at a specific location is then $4.2 \times 10^{15}\,\mathrm{molec/cm^2}$. As discussed above, the top one cm of snow as measured by Pratt et al. (2013) contains approximately $6 \times 10^{15}\,\mathrm{molec/cm^2}$. One cm of sea ice can hold, again with the measurements of Pratt et al. (2013), about $9 \times 10^{16}\,\mathrm{molec/cm^2}$, more than one order of magnitude larger than the upper limit of bromide released in the model. Upward migration of sea salt from sea ice is thus indeed a practically unlimited bromide source for modeling purposes, assuming the values found by Pratt et al. (2013) and the median and higher values measured by Krnavek et al. (2012). However, it is likely that upward migration is only effective for

sufficiently shallow snowpacks, which was suggested to be approximately 17 cm by Domine et al. (2004). So the assumption of an unlimited bromide reservoir could be unproblematic for shallow snowpacks, but is likely to be incorrect for deeper snowpacks. Most of the initial concentration of bromide is consumed by early March, so the majority of bromine emitted comes from the replenishment.

Any emissions without deposition of gas-phase species, e.g., from the sunlit condensed-phase (Pratt et al., 2013; Halfacre
et al., 2019), oceanic emissions of brominated species and emissions due to blowing snow (Yang et al., 2008, 2019) are currently not considered. The bromine emission in the model is active independent of temperature. In the Arctic ocean, temperatures are below $0°C$ throughout the simulations and even at Hudson or Baffin Bay, temperatures are below $0°C$ with few exceptions.

## 3 Retrieval of the tropospheric BrO VCD from TROPOMI observations

In this section, the algorithm to retrieve tropospheric BrO vertical column densities is presented and possible sources of bias in
the measurements are discussed.

### 3.1 Description of the retrieval algorithm

Localized BrO events can be seen best from space using data from the TROPOMI (Veefkind et al., 2012) onboard ESA's Sentinel-5P satellite. On one hand the instrument combines a high signal-to-noise ratio with an unprecedented spatial resolution of 3.5x7 $km^2$ at nadir (improved to 3.5x5.5 $km^2$ in August 2019), on the other hand its swath width of approximately 2600 km
allows for a complete coverage of polar regions several times during one day.

To retrieve tropospheric BrO from TROPOMI measurements by means of Differential Optical Absorption Spectroscopy (DOAS; Platt and Stutz, 2008), the typical DOAS approach is followed: First, the BrO concentration integrated along the light path, the so-called slant column density (SCD), is derived from the spectrum and then the vertical column density (VCD) is calculated from the SCD.

In the first step, the SCDs are retrieved using the universal DOAS fit routine from Borger et al. (2020) together with the fit settings described in Tab. 2. Next, the stratospheric fraction of the BrO SCD is separated from the tropospheric fraction. The stratospheric BrO SCD is estimated using the simultaneously retrieved columns of ozone and $NO_2$. For this purpose the correlation of these columns with stratospheric dynamics and BrO chemistry and subsequently filter pixels deemed tropospheric by a statistical data analysis are utilized. The tropospheric BrO SCD can then be obtained via $\mathrm{SCD_{trop} = SCD_{total} - SCD_{strat}}$.
One important advantage of this approach is that it does not involve model data and thus can resolve stratospheric BrO patterns on the much higher resolution of the TROPOMI instrument. For a more detailed description of the theoretical basis of this algorithm see Sihler et al. (2012).

In a second step, the obtained tropospheric BrO SCD is converted into a tropospheric BrO VCD with the help of the tropospheric **A**ir **M**ass **F**actor (AMF): $\mathrm{VCD_{trop} = SCD_{trop}/AMF_{trop}}$. The AMF is derived from retrieved $O_4$ SCDs and top
of atmosphere reflectances at 372 nm, acting as proxies for cloud coverage and surface albedo, respectively, and using lookup tables of radiative transfer simulations from McArtim (Deutschmann et al., 2011).

**Table 2.** DOAS fit settings

| Parameter | BrO Fit | O$_4$ Fit |
|---|---|---|
| Fit window | 336-360 nm | 355-390 nm |
| Absorption cross sections | BrO, 223 K (Fleischmann et al., 2004) <br> O$_4$, 203 K (Thalman and Volkamer, 2013) <br> O$_3$, 223 K, 243 K (Serdyuchenko et al., 2014) <br> NO$_2$, 220 K (Vandaele et al., 1998) <br> SO$_2$, 203 K (Bogumil et al., 2003) <br> OClO, 293 K (Bogumil et al., 2003) | O$_4$, 293 K (Thalman and Volkamer, 2013) <br> O$_3$, 243 K (Serdyuchenko et al., 2014) <br> NO$_2$, 220 K (Vandaele et al., 1998) |
| Ring effect | Two Ring spectra calculated from daily irradiance | Two Ring spectra calculated from daily irradiance |
| Polynomial | 5$^{\text{th}}$ order | 4$^{\text{th}}$ order |
| Pseudo-absorbers | inverse spectrum <br> shift and stretch <br> 2 x Pukite O$_3$ terms (Puķīte et al., 2010) at 223 K | inverse spectrum <br> shift and stretch |

As a last step, all measurements are assessed regarding their sensitivity to near surface concentrations using the measured reflectances and O$_4$ SCDs based on the modeled radiative transfer scenarios. A measurement is decided to be possibly insensitive if the measured proxies yield a tropospheric AMF below a certain sensitivity threshold in the corresponding RT simulations. Measurements deemed to be possibly obscured are discarded for all following investigations. This is discussed in more detail by Sihler et al. (2012).

### 3.2 Uncertainties and biases of the measurements

The main sources of uncertainty for the tropospheric BrO VCD are the retrieved stratospheric BrO slant column densities and the AMF used for the conversion of the SCDs to VCDs. This will be discussed both in order, starting with the possible biases in the stratospheric SCDs.

This retrieval is suitable to derive an empirical error estimation for the stratospheric SCD column. The propagated relative uncertainty for the tropospheric SCDs can be estimated to be around 10-15 %, with a lower limit for certainty of detection at around $10^{13}$ molec cm$^{-2}$. With the lowest AMF values around 0.5, this gives us a detection limit of approximately $2\times10^{13}$ molec cm$^{-2}$ for the tropospheric VCD, below which we cannot be certain that noise is not the dominating source of the signal.

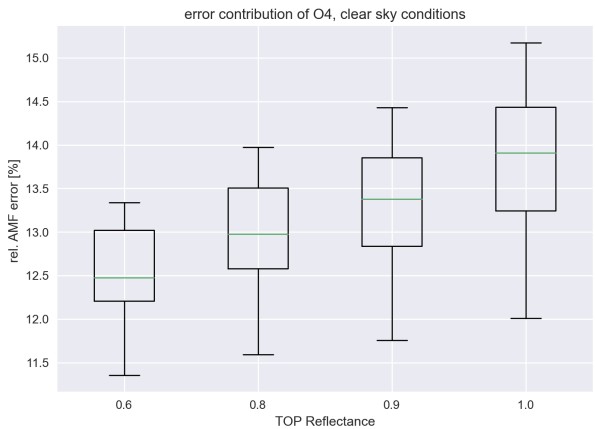
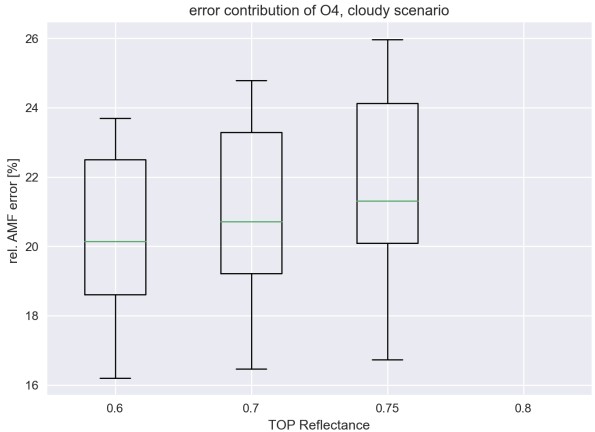

**Figure 2.** Contribution of the $O_4$ uncertainty to the overall AMF error for different reflectances. **Columns**: Left, clear sky scenario (high $O_4$ AMF). Right, cloudy scenarios (low $O_4$ AMF).

The main source of uncertainty for air mass factors in a parameter range deemed to be sensitive to near surface concentrations is the uncertainty in the measured $O_4$ SCDs which are used to calculate the AMFs from LUTs. As the relative error of the $O_4$ AMF is below 7.5 %, we can estimate the influence of the $O_4$ AMF error on the tropospheric AMF for different viewing geometries, reflectances and cloud scenarios. This yields an estimate for the relative uncertainty in the AMF of around 10-
15 % for the clear sky scenario and 15-25 % for a mostly cloudy scenario as depicted in Fig. 2. Note the missing values for a reflectance of 0.8 for the cloudy scenario, since the surface sensitivity filter considers these scenarios possibly too clouded to be sensitive to the surface(high reflectance together with low visibility of $O_4$). Other scenarios can have much higher relative errors but the corresponding measurements are discarded anyway as they are deemed to be possibly obscured.

Combining both sources of error gives an estimate of 15-30 % for the relative uncertainty of the measured troposheric BrO
VCDs.

The presence of clouds might impede the retrieval depending on the thickness of the cloud. However, especially over bright surfaces (high albedo), which are typical for polar regions, a substantial enough fraction of the observed photons will have penetrated near-surface layers even for cloudy scenarios as our radiative transfer simulations have shown. Only thick or very extended clouds make it very unlikely to observe photons from near-surface layers. These measurements are therefore discarded
based on the measured $O_4$ absorption. Note also that this surface sensitivity filter is designed to have a very small false positive rate and discards a lot of measurements which are nevertheless sensitive to near surface absorber in order to achieve this. The only limitation of this approach is the dependence on the cloud top height, as the shielding of very low and thick clouds might be underestimated if O4 is still abundant enough above this cloud. This can be safely assumed to occur very rarely in reality though, allowing us to presume that the retrieved measurements are therefore also sensitive to near surface absorbers. A more
in-depth discussion of possible biases can be found in a study by Sihler et al. (2012).

## 4 Results and discussion

In this section, model results are compared to in-situ measurements, satellite observations, and ozone sonde flights. An overview of the four simulations conducted in this work is given in Tab. 1. The first three simulations are initiated on February 1, and the high bromide simulation is started on April 1, 2019 using the results of the medium bromide simulation as initial conditions. Since the bromide replenishment time scale is set to one week, major differences between the medium and high bromide simulation are to be expected after April 8.

### 4.1 In-situ measurements

The modeled ozone mixing ratio is compared to in-situ measurements at Utqiaġvik (Alaska), Summit (Greenland), Eureka (Canada) (McClure-Begley et al., 2014), Nord Station (Greenland), Pallas (Finland) and Zeppelin Mountain (Svalbard) (Tørseth et al., 2012), see Fig. 3 for all three simulated months and Fig. 4 for April only. Statistics for the modeled ozone mixing ratio in comparison to in-situ measurements at the various locations are shown in Tab. 3 for the complete three month period and in Tab. 4 for April only. Listed are the correlation $R$, simulated average ozone mixing ration, mean bias, MB and root mean square error, RMSE. For a positive value of the mean bias, the average of the modeled ozone mixing ratio is larger than the average of the observed ozone mixing ratio.

One motivation of introducing the finite bromide assumption is to better match simulation and measurement of the ozone mixing ratio at Utqiaġvik. During February and March, the infinite FY ice bromide simulation agrees quite well with the observations, except for an overestimated ozone depletion around March 16, see Fig. 3. In April, however, the infinite FY ice bromide assumption leads to a full ozone depletion for nearly the whole month, whereas the observations find only partial to small ozone depletion events for most of the month. In other studies using the same bromine emission mechanisms, overestimation of ozone depletion also occurs frequently during March and April: Toyota et al. (2011), Falk and Sinnhuber (2018), Herrmann et al. (2021) and Marelle et al. (2021) found several instances of overestimation of ozone depletion at locations such as Alert (Canada), Utqiaġvik, Nord Station and Zeppelin Mountain. This may be due to readily available bromide depleting over time, so that the infinite FY ice bromide assumption becomes invalid. All simulations using a finite bromide assumption improve the agreement with the observations significantly, including the weak ozone depletion around March 16. The correlation increases from 0.66 to 0.81 for the low bromide simulation, see Tab. 3. The good agreement with the observations suggests that the parameters of releasable bromide and the replenishment timescale of the bromide over FY ice are reasonably chosen. The assumptions of the medium bromide simulation allow for emissions from land near Utqiaġvik and cause a slight overestimation of ozone depletion (the mean bias decreases from 1.1 to -1.8 nmol/mol) and a decrease in correlation coefficient from 0.81 to 0.79. In April, the three finite bromide simulations have nearly the same correlation of 0.75, but the mean bias and the RMSE worsen for the medium and high bromide simulations indicating that the initially releasable bromide on land-covered snow may be chosen too high for the medium and high bromide simulations.

At Summit, the model finds very little influence of the halogen chemistry on ODEs. The measurements show no obvious ODEs. The correlation of 0.3 is relatively low. Ozone is generally larger in the model, especially during March, when a very

long, weak partial ODE might have occurred. However, it is more likely that dynamical errors cause the discrepancy in March.
During a measurement campaign in 2007 and 2008 (Stutz et al., 2011; Liao et al., 2011), small BrO mixing ratios of around 2 and 5 ppt were measured, which is consistent with the modeled BrO. On a few days, such as March 22 and April 5, modeled BrO mixing ratios exceed 5 ppt and reach up to 30 ppt. Significant bromine emissions over Greenland might improve the results during March, however, there is evidence against large bromine emissions over Greenland. Therefore, Greenland is effectively excluded from emitting new bromine in the model due to the height dependence of Eq. (1), which was included due to a lack
of evidence for BrO over Greenland. The satellite data, see Fig. 9, do not show any enhanced BrO over Greenland except at the very edges of the island. The in-situ measurements at Summit also suggest small bromine emissions, if any at all. Additionally, assuming that the sea salt originates from sea water, sea salt may not easily be transported to higher elevations. In fact, there might be an overestimation of the orographic lift towards the interior of Greenland since the southern part of Greenland is part of domain 1 with a 60 km resolution only, so that the topography is smoothed and might allow air masses in the model to be
blown over Greenland, which in reality would have stayed near the sea ice.

    A further motivation of introducing the finite bromide assumption was to improve the model results in Eureka. Bromide emissions on MY are likely smaller than on FY ice, which is difficult to implement with an assumption of infinite bromide. One possibility is to directly modify the emission rate on MY ice, which would correspond to reduce the deposition velocity. The finite bromide assumption allows for a more natural differentiation of MY and FY ice. Eureka is surrounded by MY ice and
395 snow covered islands, which are not allowed to emit new bromide with the infinite FY ice bromide assumption. Probably due to the lack of nearby bromide sources, bromine emissions are strongly underestimated, which results in a consistent overestimation of ozone levels. With the finite bromide assumption, weak bromine emissions are allowed on MY ice and snow covered ice. However, the prediction of ozone has not changed much between the infinite and low bromide simulation and actually becomes a little worse during April for the low bromide assumption. This suggests that the real releasable bromide levels for
MY ice and/or snow covered land are probably significantly higher than those assumed for the low bromide simulation. The medium finite bromide simulation improves the results at Eureka even over the infinite FY ice bromide simulation, but still not removes the underestimation of ODEs. Increasing the emissions on land and MY ice as part of the high bromide simulation

**Table 3.** Statistics for the modeled ozone mixing ratio in comparison to in-situ measurements at different locations from February 1 to May 1, 2019. The three values in each entry are for the three simulations in this order: Infinite FY ice bromide, low bromide, medium bromide. The best value is marked in **bold**.

| location | R [-] | simulated average [nmol/mol] | MB [nmol/mol] | RMSE [nmol/mol] |
|---|---|---|---|---|
| Utqiaġvik | 0.66, **0.81**, 0.79 | 21.4, 29.1, 26.2 | -6.5, **1.1**, -1.8 | 13.3, **7.2**, 8.4 |
| Summit | 0.30, 0.32, **0.34** | 49.5, 49.3, 49.3 | 6.3, 6.2, **6.1** | 11.2, 10.7, **10.7** |
| Eureka | 0.51, 0.47, **0.58** | 29.8, 32.1, 28.8 | 6.4, 8.7, **5.4** | 14.0, 15.3, **12.9** |
| Nord Station | 0.64, 0.55, **0.65** | 36.4, 38.2, 36.2 | 2.1, 2.0, **0.6** | 8.5, 9.2, **8.2** |
| Pallas | 0.71, **0.88**, 0.83 | 33.1, 38.5, 35.0 | -10.1, **-4.7**, -8.1 | 11.9, **6.3**, 9.6 |
| Zeppelin Mountain | 0.48, 0.62, **0.67** | 29.4, 33.1, 31.4 | -10.3, **-6.6**, -8.4 | 14.0, **9.7**, 10.9 |

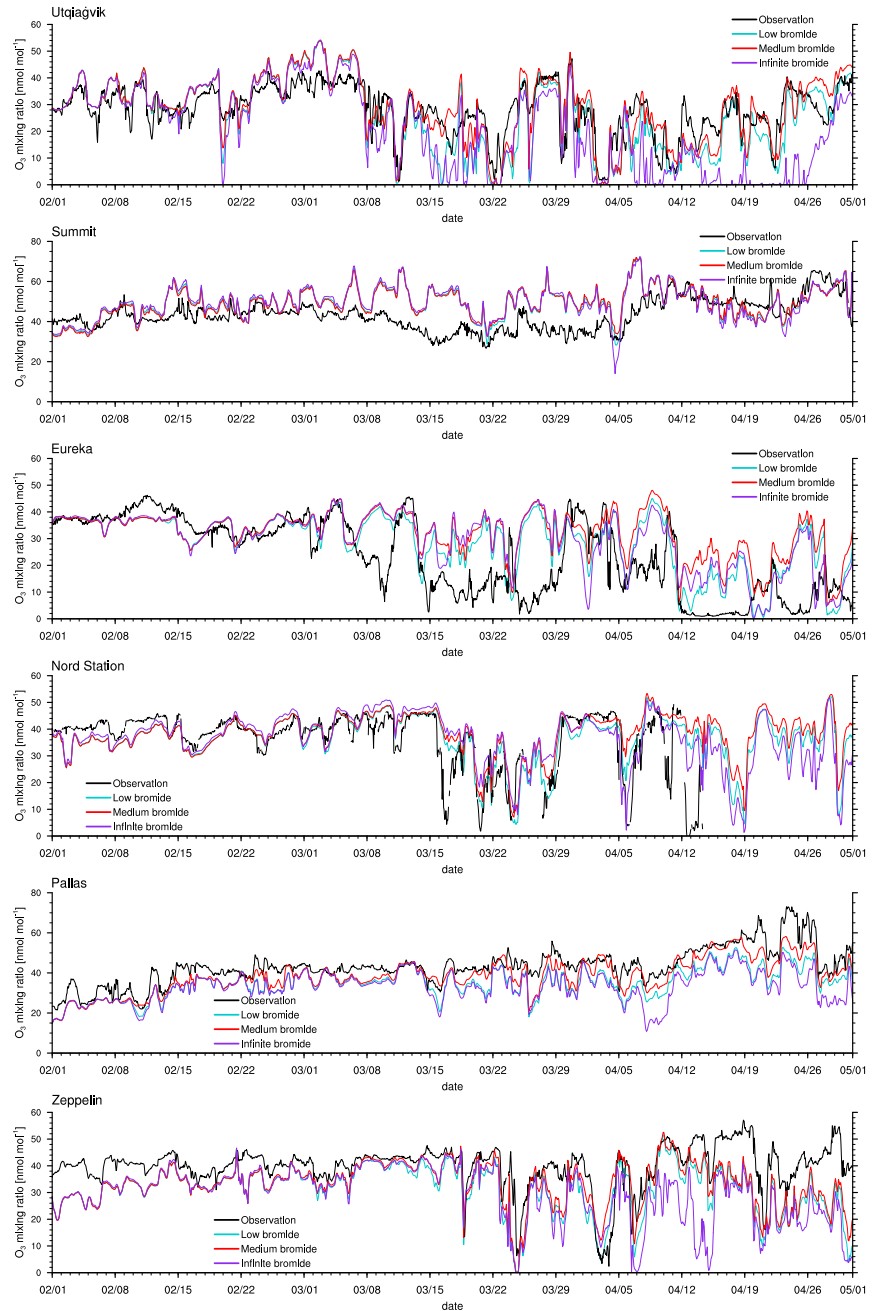

**Figure 3.** Modeled and observed ozone mixing ratios at Utqiaġvik, Summit, Eureka, Nord Station, Pallas, and Zeppelin Mountain from top to bottom.

further improves the statistics, but is still not able to significantly improve the ODEs in the second week of April. Around April 25, all four simulation results differ only little, so that it is unlikely that increasing emissions even further will cause significant

changes. Some bromine source or another ozone depletion mechanism might be missing to explain the ODEs in Eureka, such as blowing snow, which was discussed by Marelle et al. (2021), who showed some ODEs that are not explainable with emissions from the snow surface alone. It is, however, more likely that the model resolution is not sufficient to correctly simulate ODEs at Eureka since the topography around Eureka is very complex. The weather station at Eureka is located at a height of 10 m, whereas the closest grid cell is at an altitude of 155 m. Additionally, it is possible that a significant amount of bromine affecting the chemistry at Eureka is emitted from snow-covered sea ice at the strait and fjords near Eureka, which are less than 20 km wide and can thus not be resolved by the model. Comparing the results of domain 2 with 20 km resolution to those of domain 1 with a 60 km resolution reveals that ODEs at Eureka are affected by the resolution: The correlation of model results with observations at Eureka for the refined domain 2 is, in comparison to domain 1, increased by 0.02, 0.09 and 0.08 for the infinite, low and medium bromide simulations, respectively, still remaining at overall relatively low values of the correlation coefficient below about 0.6. Differently to Eureka, the correlations of the simulations with the two different domains with observations at Utqiaġvik agree within 0.01 for all simulations, which is likely due to the simpler local topography at Utqiaġvik.

At Nord Station, the observations are not available at the end of April. The comparison to the simulations between February 1 and April 15 shows some similarities to Eureka, in that ODEs are generally underestimated, however, the agreement between simulations and observations at North Station is better compared to Eureka. Nord Station is surrounded by mostly MY ice and snow covered land, so that a similar discussion as at Eureka for the underestimation at North Station holds. Also similar to Eureka, the medium finite bromide simulation performs slightly better than the infinite FY ice bromide simulation and the simulation results improve with larger emissions, with a correlation of 0.56, 0.62 and 0.7 for the low, medium and high bromide simulations, respectively. The weather station is located at an altitude of 20 m, whereas the closest grid cell is at 85 m. Differently to Eureka, however, there is no improvement for the increased grid resolution of domain 2. Correlations of results of domain 2 with observations increase by 0.04, -0.02 and -0.03 in comparison to domain 1 for the infinite, low and medium bromide simulations, respectively.

At Pallas, little ozone depletion is observed except for potential ODEs on April 22 and from April 27 to May 1, whereas the model finds a few additional partial ODEs at the end of March and in April. The source of bromine is most likely the nearby White Sea located at a distance of about 500 km. Due to the reduced bromine sources, the low bromide simulation shows the

**Table 4.** Statistics for the ozone mixing ratio at different locations from April 1 to May 1, 2019. The four values in each entry are for the four simulations in this order: Infinite FY ice bromide, low bromide, medium bromide, high bromide. The best value is marked in **bold**.

| location | R [-] | sim. average [nmol/mol] | MB [nmol/mol] | RMSE [nmol/mol] |
|---|---|---|---|---|
| Utqiaġvik | 0.35, 0.75, **0.76**, 0.75 | 8.0, 23.5, 19.0, 16.4 | -15.7, **-0.2**, -4.7, -7.3 | 19.1, **7.8**, 9.3, 10.3 |
| Summit | **0.44**, 0.35, 0.39, 0.39 | 50.1, 51.7, 51.2, 51.0 | **1.1**, 2.6, 2.1, 2.0 | 9.5, 9.5, **9.3**, 9.4 |
| Eureka | 0.31, 0.41, 0.45, **0.49** | 20.1, 26.8, 20.5, 17.8 | 7.5, 14.2, 7.9, **5.2** | 14.9, 18.5, 14.6, **12.9** |
| Nord Station | 0.65, 0.56, 0.62, **0.70** | 32.0, 40.5, 36.7, 33.5 | **4.0**, 10.7, 7.4, 5.4 | **12.6**, 17.3, 14.7, 12.9 |
| Pallas | 0.74, **0.83**, 0.79, 0.80 | 34.7, 44.9, 39.3, 39.0 | -15.5, **-5.2**, -10.9, -11.2 | 16.8, **7.3**, 12.2, 12.4 |
| Zeppelin Mt. | 0.47, 0.58, **0.67**, 0.65 | 22.0, 31.6, 28.4, 26.9 | -17.6, **-8.0**, -11.1, -12.7 | 20.9, **12.9**, 14.5, 15.9 |

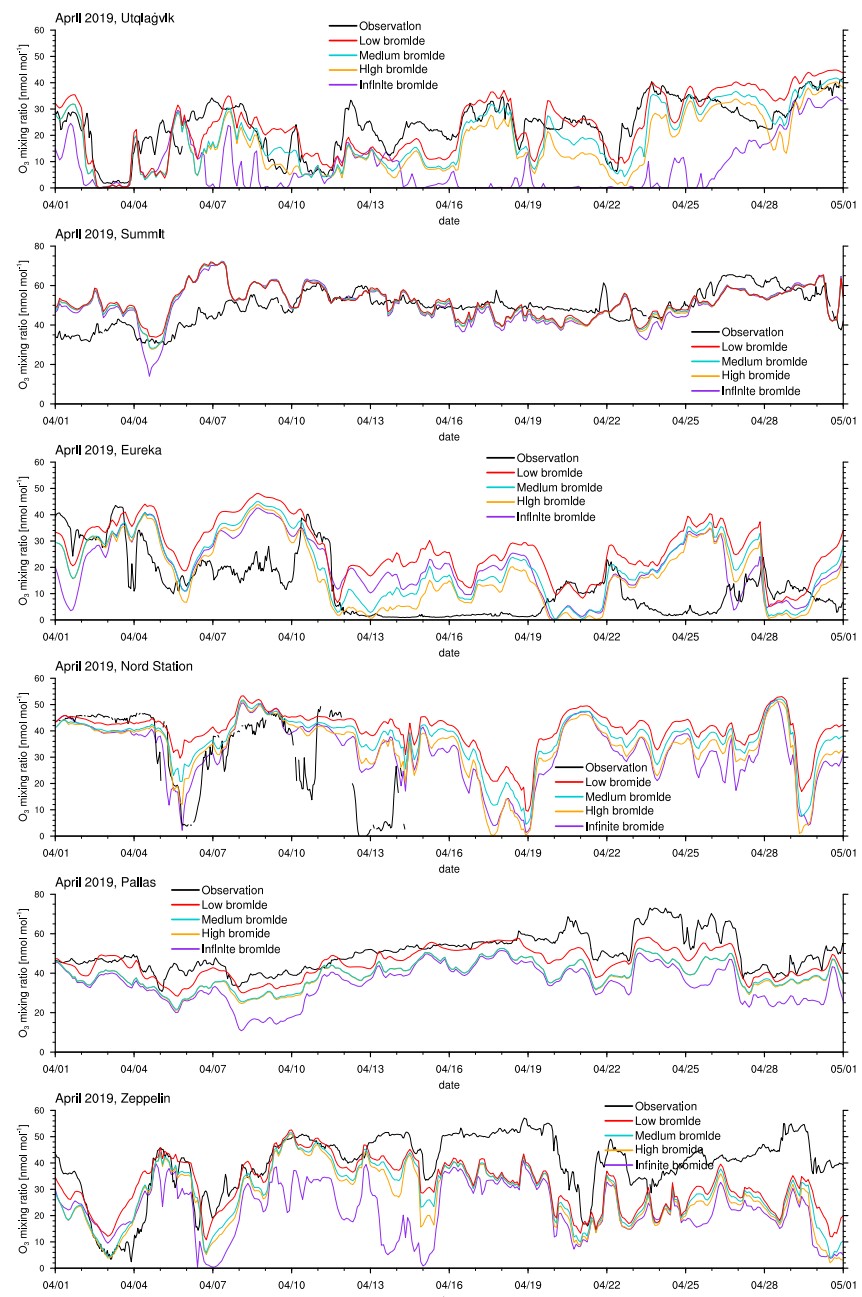

**Figure 4.** Modeled and observed ozone during April at Utqiaġvik, Summit, Nord Station, Pallas, and Zeppelin Mountain from top to bottom.

best agreement with the observations and a high correlation of 0.88. There is surprisingly little difference between the medium and high bromide cases, which suggests that emissions mostly come from the FY ice. The changed definition of snow-covered land, which allows for recycling of bromine near ice-covered coast water, is likely to cause the differences.

Finally, Zeppelin Mountain is located in an area close to FY ice, which again causes an overestimation of ODEs in April for the infinite FY ice bromide simulation. The finite bromide simulations show an improved agreement with the observations, but ozone is still underestimated during April. Ozone levels around March 19 are similar for all simulations, which suggests that the bromide emission mechanism is not the cause of the underestimation. Even though the correlation of the low bromide simulation and the observations are slightly worse compared to the medium and high bromide simulations, the low bromide simulations performs best which shows the lowest mean bias and RMSE. Here, the weather station is located at an altitude of 475 m, with the closest grid cell at 245 ms. There is a clear improvement of the results at Zeppelin Mt. with a higher grid resolution: Correlations of simulation results of domain 2 with observations increase by 0.12, 0.11 and 0.09 for the infinite, low and medium bromide simulations, respectively, in comparison to the coarser domain 1.

Overall, the low bromide simulation performs best for the measurement sites near FY ice and the high bromide simulation is best at Eureka and Station Nord, which is located near MY ice. This suggests that MY ice might indeed be effective, albeit not necessarily as effective in emitting bromine as FY ice, whereas larger emissions for snow-covered land generally reduce the agreement with the observations.

## 4.2 Vertical profiles of $O_3$ and BrO

The modeled ozone mixing ratio and potential temperature profile are compared to ozone sonde measurements at Churchill, Alert, Eureka, and Resolute (Meteorological Service of Canada, 2021), all located in Canada, see Figs. 5-8. The infinite FY ice bromide, low bromide and medium bromide are shown on the left, center, and right columns, respectively. The rows display results at different dates denoted in the figure caption. Figure 5 shows numerical results compared to observations from sonde flights at Churchill. While bromine release and ozone depletion at Churchill, which is located at a relatively low latitude of 58.74°N, are strong in February (not shown) and early March, they are typically weaker during April.

On March 2, the models underestimate the ODE, except for the medium bromide simulation, which agrees very well with the measurement. On February 6 and 15, modeled ozone levels are similar to observed levels (not shown). On all other days, February 24, March 13 and 20 and April 3, 10, 17, and 24 (not shown), ozone depletion is overestimated by the models, where the low bromide simulation performs best, followed by the medium and the high bromide simulation. On most days, the infinite FY ice bromide simulation shows the largest discrepancies to the observations. On March 20 and April 3, the low bromide simulation agrees very well with the observations.

In summary, ozone depletion is generally overestimated near Churchill, which will also be evident in comparison to the satellite data, see section 4.3. The model generally overestimates BrO VCDs around Hudson Bay. The releasable bromide and the replenishment timescale are chosen globally in this study, but perhaps on the FY ice at Hudson Bay, the releasable bromide is smaller or the replenishment timescale is larger in reality. Adding the emissions on snow-covered land reduces the agreement with the observations. Since the medium bromide simulation shows the best agreement on March 2, a combination of reduced emissions on FY ice at Hudson Bay and small emissions on snow-covered lands nearby might lead to the best agreement with observations. Smaller releasable bromide values at Hudson Bay could be explained by the low salinity of the sea water (Myers

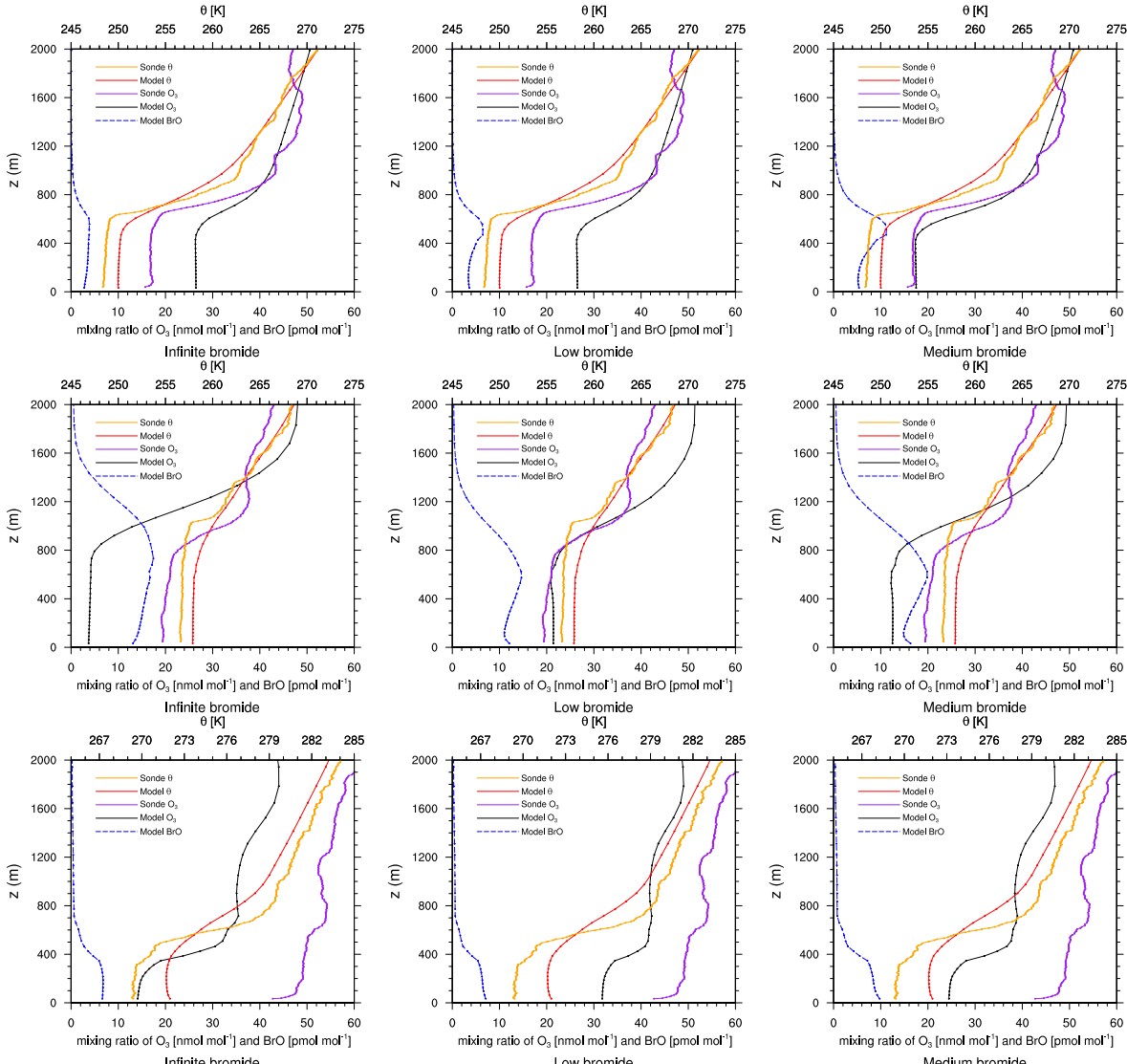

**Figure 5.** Modeled BrO mixing ratio as well as potential temperature $\theta$ and ozone mixing ratios from simulations and from ozone sonde flights against height at Churchill. **Columns**: Left, infinite FY ice bromide. Center, low bromide. Right, medium bromide. **Rows**: Top, March 2, 2019. Center, April 3, 2019. Bottom, April 17, 2019

et al., 1990; Jones and Anderson, 1994), which is due to a high influx of fresh water into the approximately 125 m deep bay, the low rate of evaporation due to the sea ice cover, and the slow exchange of water with the Arctic ocean.

Results of ozone sonde flights and numerical simulations at Alert are shown in Fig. 6. The models and observations agree well on February 6, 13 and March 6 and 13 (not shown), however, no ODE occurs at those days. On April 3, an ODE near the ground is found by all simulations and the observations, however, there seems to be a large difference in the boundary layer

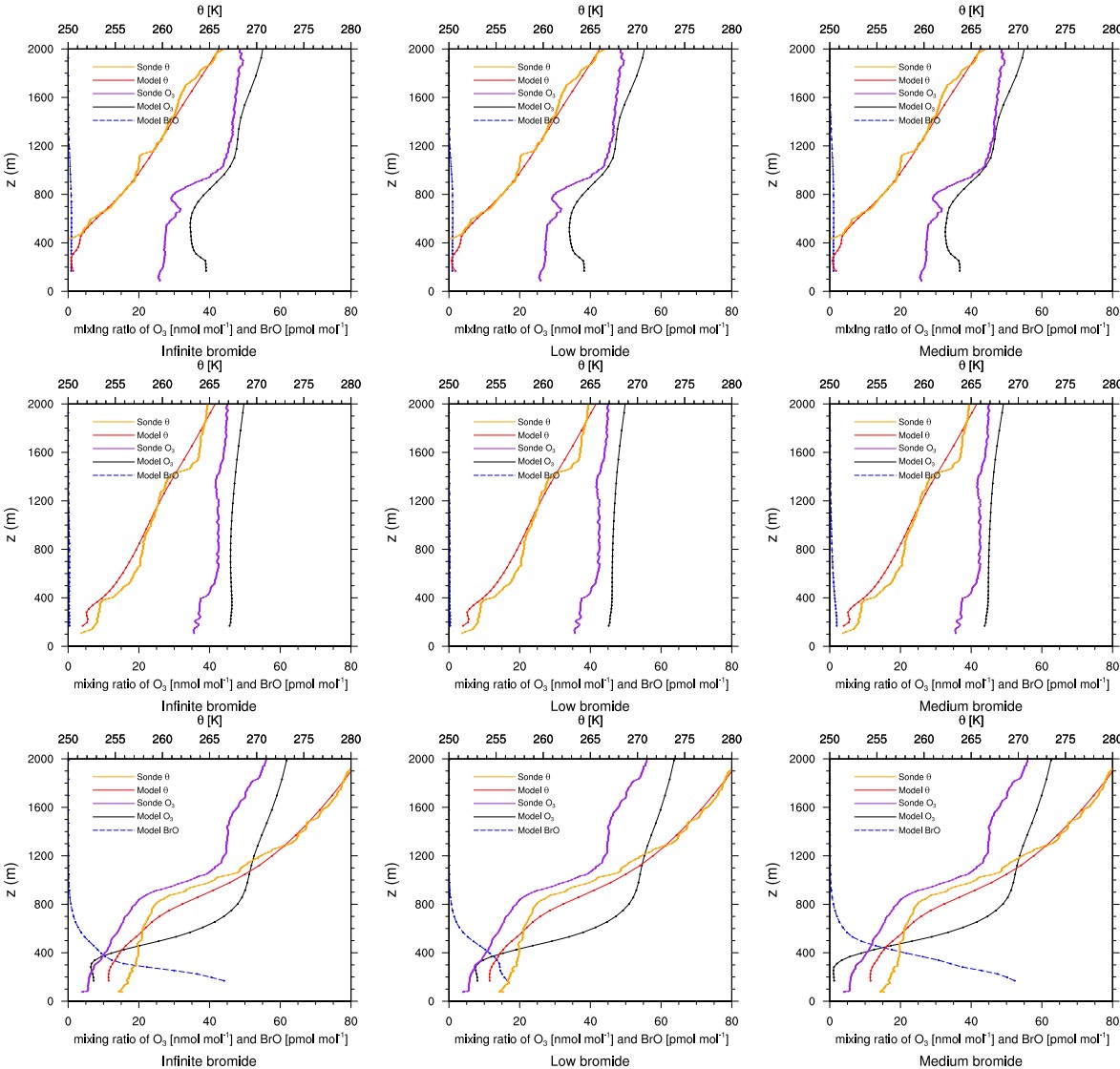

**Figure 6.** Modeled BrO mixing ratio as well as potential temperature $\theta$ and ozone mixing ratios from simulations and from ozone sonde flights against height at Alert. **Columns**: Left, infinite FY ice bromide. Center, low bromide. Right, medium bromide. **Rows**: Top, March 6, 2019. Center, March 13, 2019. Bottom, April 3, 2019

height, which may be due to the difference in heights at the nearest grid cell of 169 m and the observation site, 75 m. It should be noted that simulations, in general, struggle with modeling boundary layers over snow, see for example the study of Sterk et al. (2015). Both the infinite and low bromide simulations agree well on that day, whereas the medium and high bromide simulations overestimate the ODE near the ground.

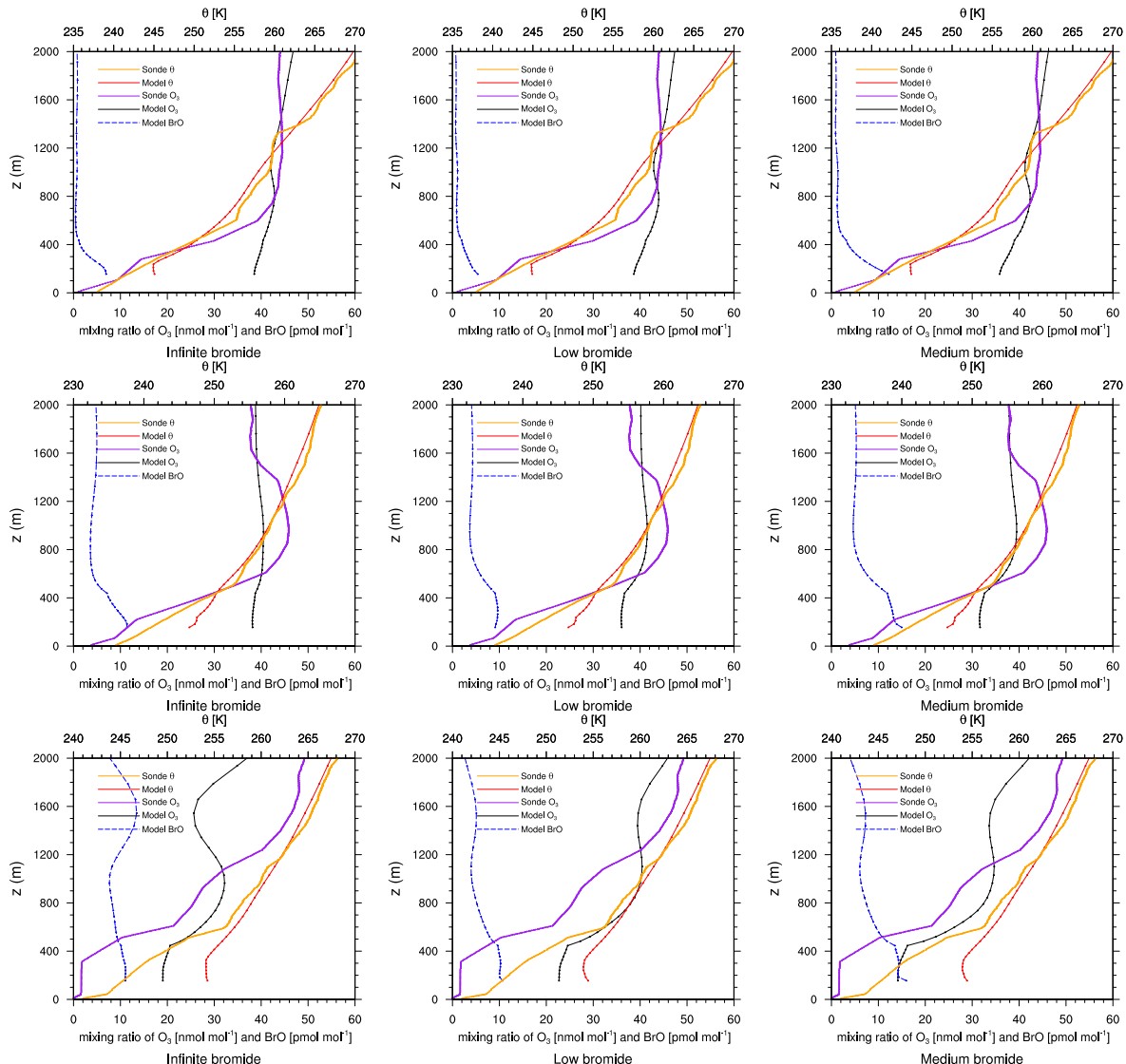

**Figure 7.** Modeled BrO mixing ratio as well as potential temperature $\theta$ and ozone mixing ratios from simulations and from ozone sonde flights against height at Eureka. **Columns**: Left, infinite FY ice bromide. Left Center, low bromide. Right center, medium bromide. Right, high bromide. **Rows**: Top, March 21 2019. Center, March 28, 2019. Bottom, April 18, 2019

Data from ozone sonde flights and simulation results at Eureka are shown in Fig. 7. The comparison to model results near the ground is consistent with what has been found for the in-situ measurements discussed in Figs. 3 and 4. There are further mismatches between measured and modeled ozone on March 8, 9, and 15 as well as on April 18 (not shown). The vertical structure of the boundary seems to be quite complex for both the simulations and the observations. Similarly to Alert, the numerical grid cell used for comparison is at an altitude of 155 m, which is at a higher altitude than the sonde flights (10 m).

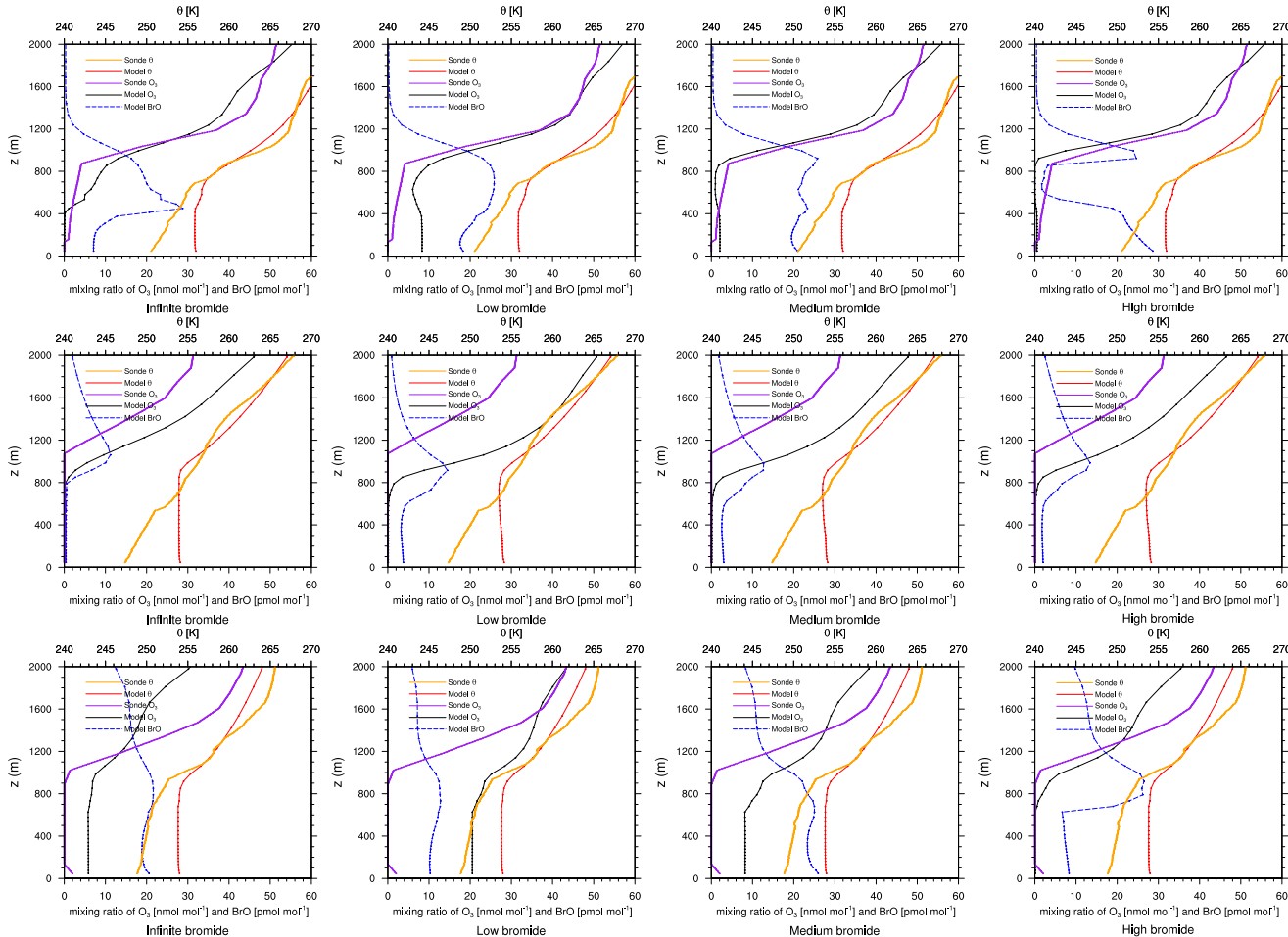

**Figure 8.** Modeled BrO mixing ratio as well as potential temperature $\theta$ and ozone mixing ratios from simulations and from ozone sonde flights against height at Resolute. **Columns**: Left, infinite FY ice bromide. Center, low bromide. Right, medium bromide. **Rows**: Top, April 4, 2019. Middle, April 18, 2019. Bottom, April 25, 2019

Thus, the discrepancies of the simulations and the observations may be caused by the resolution of the grid and the complexity of the meteorology near Eureka.

Figure 8 shows results of ozone sonde flights and numerical simulations near Resolute, Canada during April. Bromine emissions are still strong at this high latitude in April, as can be seen by the strong ODEs during the shown days. The high bromide simulation agrees best with the observations at all three days and improves the infinite FY ice bromide simulation, whereas
the low bromide simulation shows a poor performance. Resolute is located close to both MY ice and many snow-covered islands, which can emit large quantities of bromine in the high bromide simulation. In the days before April, all simulations agree well with the observations, however, the low bromide simulation slightly underestimates the ODEs in comparison to the measurements.

## 4.3 Comparison with satellite BrO VCDs

Vertical column densities of BrO are calculated by vertically integrating BrO from the ground to 4 km height. To allow comparison to the satellite data, the resulting BrO VCDs are temporally interpolated to the individual satellite orbits and averaged over one day, excluding grid points at locations possibly obstructed by the sensitivity filter in the individual satellite orbits.

The model simulates almost all of the enhanced BrO VCDs found by the satellite, four examples at different times are shown in Fig. 9. The simulated ozone mixing ratio and meteorology at the ground level for these days are shown in Figs. 10 and 11, respectively.

On February 27 displayed in top row in the Figs. 9-11, the North-South stripe-like region with the elevated BrO VCD extending over the Canadian mainland is captured by all three simulations. The BrO VCD is co-located with a meteorological front. The low bromide simulation underestimates the BrO VCD while the infinite and medium bromide simulations find VCD-values comparable to the satellite data. The simulated peak of the BrO VCD is located over the mainland for the medium bromide simulation, whereas the satellite and the infinite FY ice bromide simulation find the peak further north near the coast.

On March 19, see second row in Figs. 9-11, the large BrO VCD over Chukchi Sea and the two small stripes near the north pole are predicted by all models. The BrO VCD over Chukchi Sea is located over FY ice, where the releasable bromide values are the same for both finite bromide simulations. For that reason, the BrO VCDs simulated by the low and medium bromide simulations are very similar. The infinite FY ice bromide simulation calculates a ring-like BrO VCD over Chukchi Sea, which is caused by a full ozone depletion event over a large area covering the center of the ring and the stripe with small BrO VCDs. The finite bromide simulations simulate a full ozone depletion over a smaller area there, so that ozone from outside the ring can sustain the BrO. The structure near the north pole is also caused by a full ozone depletion, which is actually stronger for the finite bromide simulations, since the structure is located over MY ice, which is a bromide source in the finite bromide simulations.

On March 27 shown in the third row in Figs. 9-11, the models match well with the observed BrO VCD over East Siberian Sea and the main land. Interestingly, both finite bromide simulations calculate less BrO at the observed BrO VCD peak. The BrO VCD over the Canadian Arctic Archipelago is predicted by the simulations, but the low bromide simulation agrees best with respect to the magnitude. The satellite and the models find BrO VCDs at the northern coast of Greenland, however, the simulated location of the BrO resides west of the observed location.

On April 5, see the bottom row in Figs. 9-11, the models simulate both the BrO VCDs over the Canadic-Arctic Archipelago and east of the coast of Greenland. All simulations overestimate both the extent and the magnitude of the BrO VCD, where low bromide simulation is somewhat better than the medium bromide simulation which again agrees better with the observations than the infinite FY ice bromide simulation.

The model frequently simulates enhanced BrO VCDs at locations where the satellite does not observe enhanced BrO VCDs, especially in the regions around Hudson Bay and Baffin Bay. The BrO VCDs are generally weakest for the low bromide simulation, whereas the infinite FY ice bromide simulation predicts the strongest BrO VCDs on FY ice which is not observed.

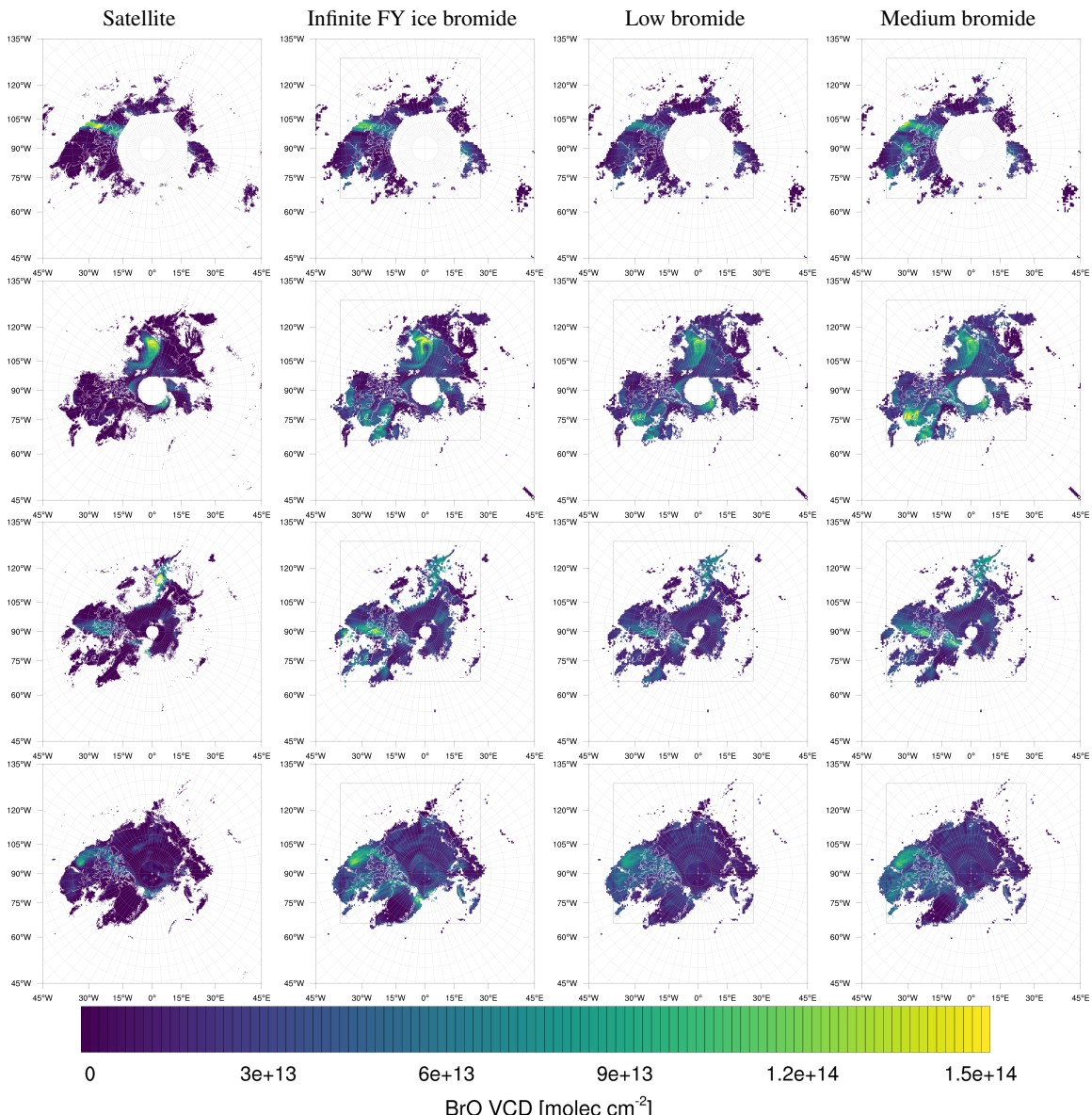

**Figure 9.** Observed (left) and simulated (infinite, low, and medium bromide from left to right) tropospheric BrO VCDs in 2019. From top to bottom: February 27; March 19; March 27; April 5

The medium bromide simulation more frequently predicts enhanced BrO VCDs not observed by the satellite on or near land, which suggests that the assumed values of releasable bromide on land are too large for the medium bromide simulation. For the finite bromine emissions, a constant replenishment of bromine is assumed, which must originate from a bromide source not explicitly modeled in the present study, such as the deeper layers of the snow, sea ice or sea water. After strong, long-

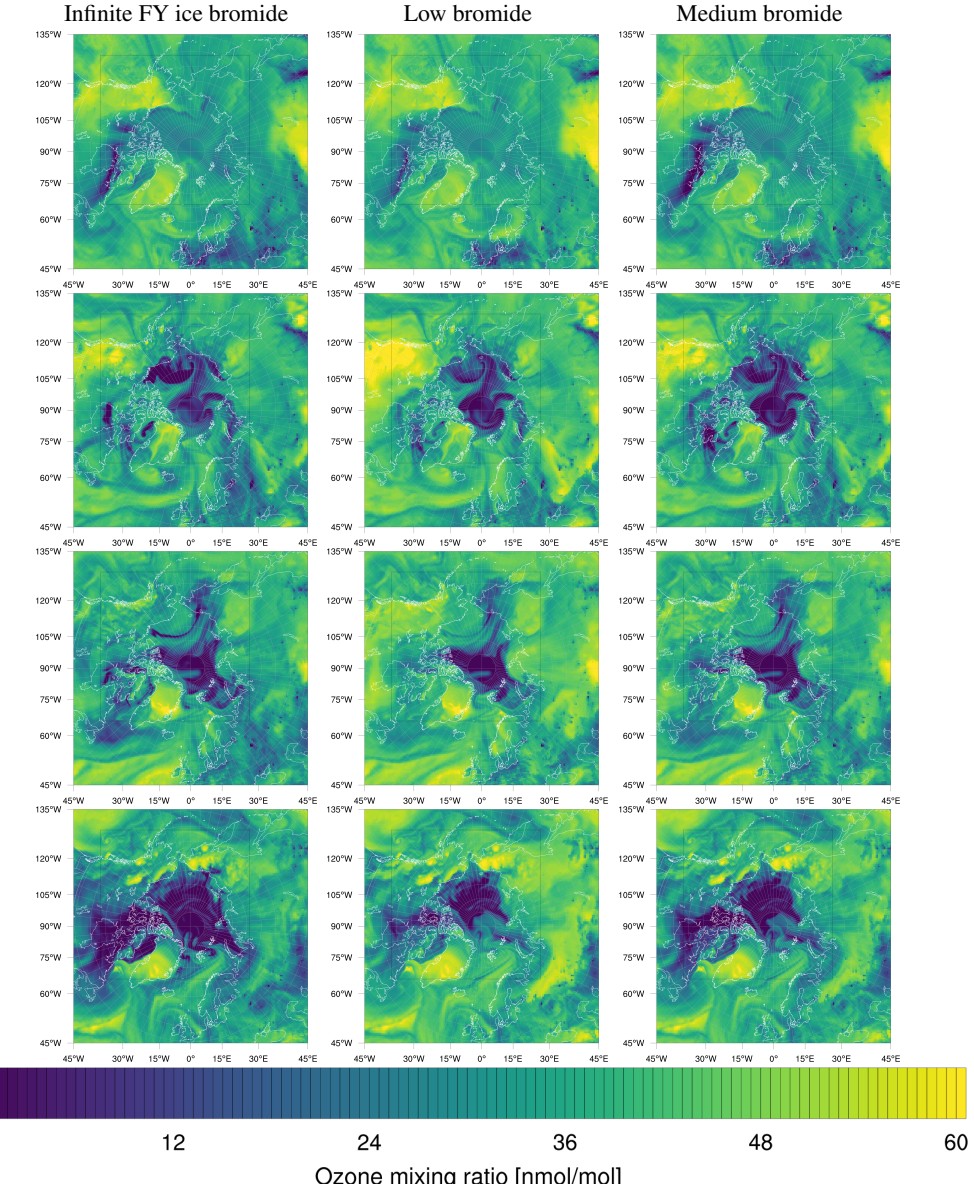

**Figure 10.** Simulated ozone mixing ratio at the ground level. From top to bottom: February 27 at 18 GMT; March 19 at 20 GMT; March 27 at 20 GMT; April 5 at 16 GMT

term bromine emissions, the constant replenishment might actually slow down at some locations like Hudson Bay, since the underlying reservoir might also be depleted. This, however, is not considered in the present model.

In Tab. 5 the temporal correlation between model and satellite pixels for different locations is summarized. These correlations were calculated by first interpolating model and satellite BrO VCDs to the same grid, then interpolating model values in time

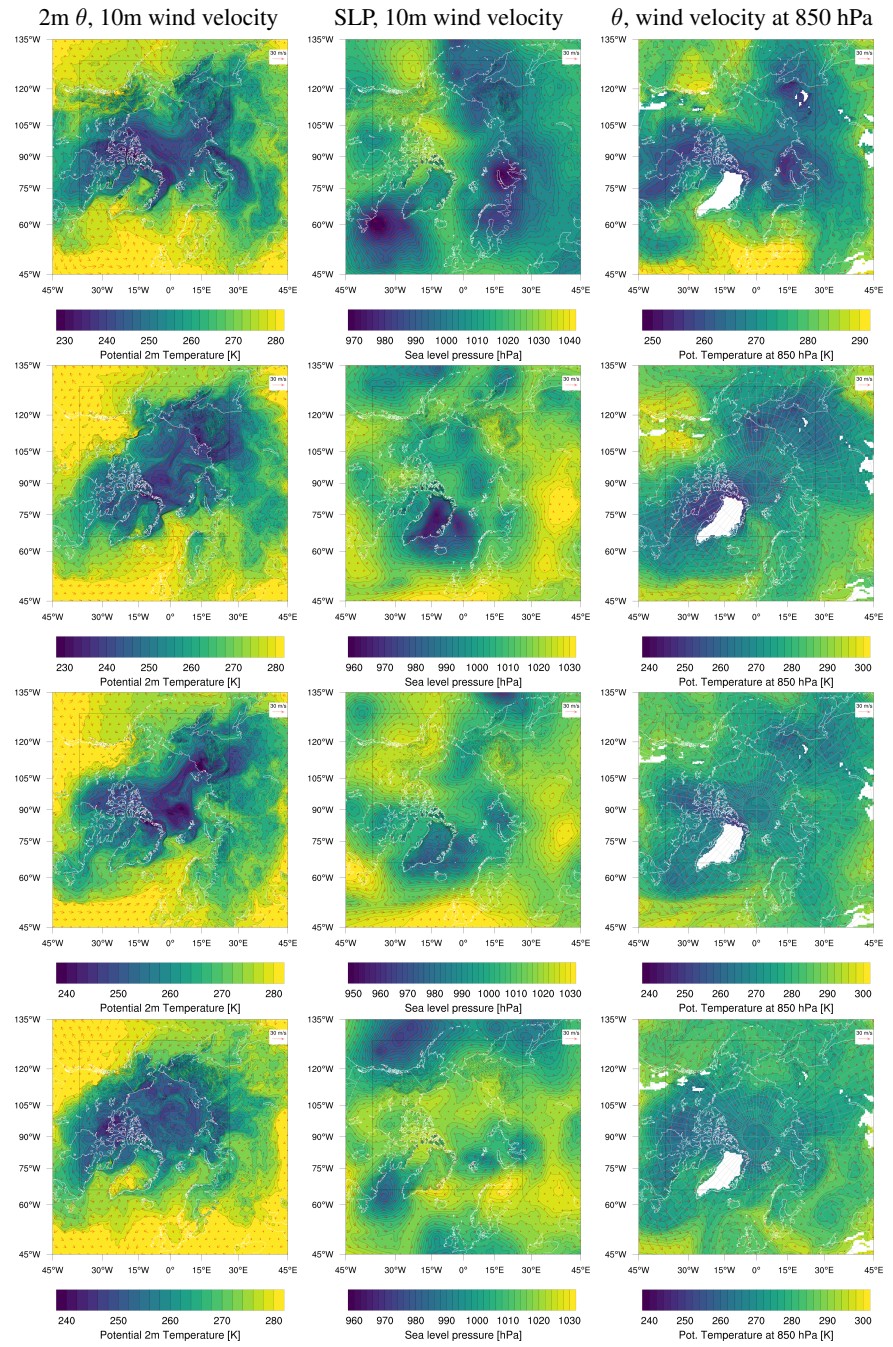

**Figure 11.** Simulated meteorology in 2019. From top to bottom: February 27 at 18 GMT; March 19 at 20 GMT; March 27 at 20 GMT; April 5 at 16 GMT

to satellite time points and calculating the correlation in time for every grid point. Finally, we took the nearest neighbor to the measurement sites in space to receive the correlation at that location. For most stations, the low bromide simulation has the highest correlation to the measurements, agreeing with the observations discussed previously. The results generally underline the good agreement between model and satellite observations. However, some details are interesting to discuss.

535 For the Eureka station it was already discussed in section 4.1 that the model resolution is probably not sufficiently high due to the complex topography. Therefore, the agreement to in-situ measurements is poor and the same can be seen in comparison with satellite maps, probably explaining the low correlation at this location.

 The correlation results at Summit agree with the results from the correlation analysis of the in-situ measurements, where the measurements showed no ODEs while the model seemed to overestimate the occurrence of partial ODEs. Nevertheless,
540 the correlation here is extremely low due to almost no observed BrO in the satellite maps over Greenland, which could also indicate a systematic underestimation of BrO for high altitudes in the satellite measurements. This might be due to problems in the estimation of the AMF for higher altitudes, where $O_4$ concentrations are low.

 At Pallas the correlation is very good and lies between 0.96 and 0.99 for the different simulations, but the surface sensitivity filter of the satellite retrieval discards almost all measurements near the station. Consequently, only very few measurements
545 are taken into account for the calculation of the correlation coefficient, all of which show no BrO enhancement. Due to the underlying poor statistics, this correlation is discarded.

 On February 10, 11, 13, and 14 and on April 2, 9, and 24, the model does not predict enhanced BrO VCDs found by the satellite. In Fig. 12, four of these days are shown. For the missed events in February, the model does simulate non-zero BrO VCD at these locations, but the magnitude is underestimated. However, enhanced BrO VCD occur at a later time of these
550 days. In early February, there is only very little sunlight at these locations. The bromide oxidation due to ozone, which is the most important bromine explosion trigger in the model, is active for solar zenith angles (SZA) smaller than 85°. Perhaps this condition is too strict and the activation of bromide due to ozone may also occur at larger SZAs.

 Over Greenland, enhanced BrO VCDs are not observed by the satellite, see Fig. 9. However, the simulations calculate enhanced BrO VCDs over Greenland on several days. Greenland is explicitly excluded as a surface for both bromide emission
555 and recycling and thus, the BrO over Greenland is produced elsewhere and then advected to Greenland. The model might

**Table 5.** Correlation coefficient at different locations from February 1 to May 1, 2019 for the three different simulations in order. The best value is marked in **bold**. The correlation coefficients for Pallas station are discarded due to poor statistics.

| location | infinite FY ice bromide R [-] | low bromide R [-] | medium bromide R [-] |
|---|---|---|---|
| Utqiaġvik | 0.44 | **0.51** | 0.46 |
| Summit | 0.03 | 0.03 | **0.04** |
| Eureka | 0.16 | **0.22** | 0.14, |
| Nord Station | 0.72 | **0.78** | 0.78 |
| Pallas | - | - | - |
| Zeppelin Mt. | 0.51 | **0.62** | 0.59, |

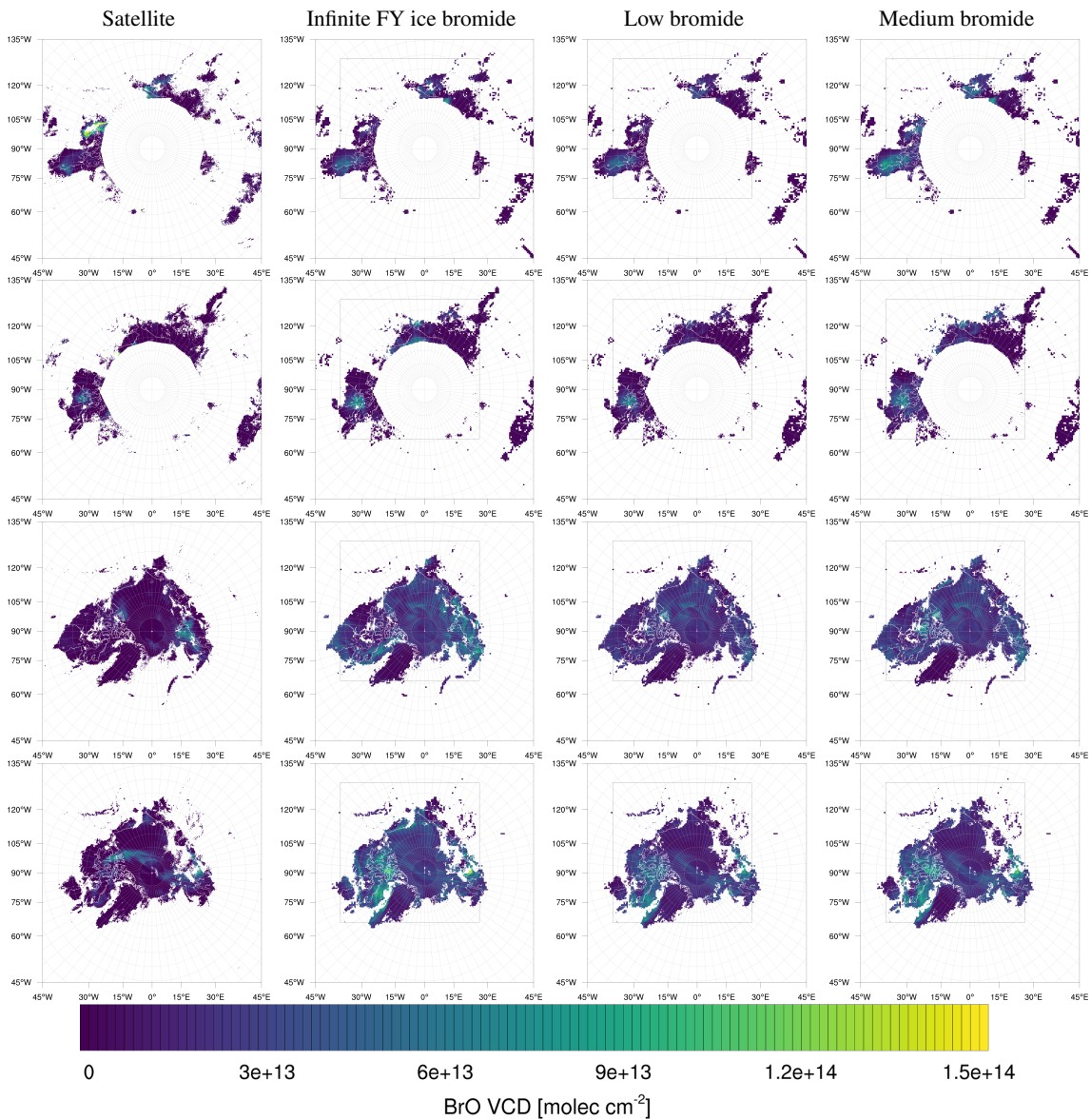

**Figure 12.** Observed (left) and simulated (infinite, low, and medium bromide from left to right) tropospheric BrO VCDs in 2019. From top to bottom: February 10, February 13, April 9, and April 24

overestimate air transport to Greenland, and perhaps the grid resolutions of 20 km and 60 km for the two domains, respectively, smoothen the topography too much. Another explanation might be the previously discussed overestimation of enhanced BrO VCDs over the regions around Hudson Bay and Baffin Bay, including the Labrador Sea, from which air is occasionally transported to Greenland.

Slightly enhanced BrO VCDs not observed by the satellite can be seen in the simulations over most of the Arctic region. The BrO VCDs are slightly reduced in the low bromide simulation compared to the other simulations, but they still occur frequently. It is unlikely that this mismatch is due to issues in the observed data. As discussed in section 3.2, it might be possible that a very low and thick cloud might effectively shield a near-surface layer of BrO. But the discussed discrepancies between model and measurements for these slighty enhanced BrO VCDs are of such frequency and extent that such a scenario
would be highly unlikely. As it was also discussed above, the lower limit of certainty for the measured BrO VCDs lies around $2x10^{13}$ molec cm$^{-2}$, whereas most of these slight BrO enhancements are a factor of two or more larger than this limit. A more likely explanation might be the bromide oxidation due to ozone, which is used as trigger mechanism and might release bromine too frequently. As long as there is both sunlight and ozone, small amounts of bromine are always released, which may explain the slightly enhanced BrO VCDs. However, no good alternative to the bromide oxidation by ozone seems to be
known. Photolytic bromine emission (Pratt et al., 2013; Wang and Pratt, 2017), possibly due to reactions with OH in the liquid phase, requires only sunlight and should thus emit even more background bromine. Blowing snow may be a possible trigger of ODEs, however, the results of Marelle et al. (2021) show little difference between purely ground-based emissions and the combination of ground-based emissions with blowing snow emissions. Marelle et al. (2021) did not explicitly test the blowing snow emissions as a replacement for the bromide oxidation by ozone, i.e., by performing a simulation with both the blowing
snow and surface emission schemes but with the bromide oxidation by ozone turned off, so that it cannot be ruled out that blowing snow might be an efficient trigger for the bromine explosion. An alternative explanation might be a missing bromine sink in the model, perhaps a weak constant sink due to fast reactions with a low-concentration species such as mercury which is currently not considered. It is of course conceivable that biases in the meteorology can result in biases of the BrO VCDs. In comparison to the ozone sonde measurements, the model underestimates the boundary layer stability, which might lead to
more BrO being released overall, but could also make it less likely to trigger a bromine explosion in the first place. However, we think it is unlikely that this would result in smaller (but still significant) BrO VCDs nearly throughout the whole Arctic. Wet removal of halogen species is currently not implemented and may act as an important sink for halogens. However, the missing wet removal for halogens is unlikely to explain all of the BrO VCDs found by the model and not by satellite, since these are consistently present throughout the computational domain, whereas precipitation affects only approximately 20 % of
grid cells at a time. Additionally, bromine levels during ODEs are only highly concentrated in the boundary layer, where dry deposition is the dominant removal process. It should be noted that only BrONO$_2$, HOBr, and HBr can be effectively removed by wet deposition, since species such as BrO are not soluble.

## 5   Conclusions

Ozone depletion events in the Arctic spring of 2019 were modeled using a modified version of the WRF-Chem 3.9.0 code (Her-
rmann et al., 2021), which was used to model ODEs in the polar spring of the year 2009. The comparison to in-situ, ozone sonde, and satellite data shows similarly good agreement to the observations for both years, showcasing the generality of the model. However, in both years, the occurrence of ODEs in April are overestimated so that an alternative to the widespread

assumption of infinite FY ice bromide content of the snow was developed which uses a new bromine emission scheme tracking the releasable bromide in snow covering the ground.

In the new scheme, releasable bromide is replenished by a constant rate proportional to its initial value to assure that bromide is not used up in later months. The amount of releasable bromide and its replenishment timescale are free model parameters. The releasable bromide is chosen to be different for FY ice, MY ice, and snow-covered land but otherwise, values are determined globally and independent of time. Three different finite bromide assumption with low, medium, and high initially releasable bromide are studied where different parameter settings on MY ice and on snow-covered land were investigated. All finite and the infinite FY ice bromide simulations show similar results in February, moderate differences in March, and large deviations in April. In comparison to the infinite FY ice bromide case, the new emission schemes greatly improve the overall agreement to the observations. The finite bromine schemes, for example, diminish the strong overestimation found with the infinite FY ice bromide scheme for ODEs at Utqiaġvik during April 2019. Furthermore, the correlation of model results and in-situ measurements at Utqiaġvik in April increase from 0.35 for the infinite FY ice bromide simulation to about 0.75 for the three finite bromide simulations. The comparison to in-situ measurements at Zeppelin Mountain and Pallas as well as to ozone sonde flights at Churchill also show significant improvement by all finite bromide simulations. At these four locations, the low bromide simulation performs best.

There are only small differences between all four simulations at Summit and Alert, mostly due to a small influence of bromine during the observed days. The high bromide simulation performs best at Eureka, Nord Station, and Resolute, followed by the medium, infinite, and low bromide simulations. This is due to the assumption that bromide emissions do not occur on MY ice and snow-covered land in the infinite FY ice bromide simulation, whereas the medium and high bromide simulations can emit large quantities of bromine on these surfaces. The medium and high bromide simulations perform slightly worse at Utqiaġvik and Pallas in comparison to the low bromide simulation, albeit still better in comparison to the infinite FY ice bromide simulation.

Almost all instances of elevated BrO VCD observed by TROPOMI are found by the simulations with a generally good agreement in shape and often best quantitative agreement for the medium bromide simulation. The low bromide simulation tends to underestimate BrO VCDs over land. All models frequently show slightly enhanced BrO VCDs throughout the Arctic not observed by the satellite. Perhaps there is a bromine sink especially effective at small bromine concentrations, for instance, a sink mechanism of zero order in the concentration of BrO and other reactive bromine species, which is not considered in the present model. Wet deposition of halogen species, which is currently not considered in the present model, may also remove part of the BrO VCDs found by the model but not by the satellite. Alternatively, the triggers of bromine explosion, primarily the bromide oxidation by ozone in the present model, might be too effective in producing initial bromine. It should be noted that the bromine explosion as an autocatalytic process has an inherently chaotic component, i.e., for example, small changes in the initial conditions may lead to large differences in the occurrence of ODEs. If the conditions are fulfilled, i.e., enough bromide and $O_3$ are available and the pH is sufficiently low, the bromine explosion may take place or not.

The medium bromide simulation is superior to the infinite FY ice bromide simulation at almost all locations and at most times, with the exception of BrO VCDs near Hudson Bay. However, the medium bromide assumption is not always better than

the low bromide simulation, especially at coastal sites such as Utqiaġvik and Pallas. Thus, no unambiguously best simulation configuration has been found in this work. The values of releasable bromide and the replenishment timescale were chosen globally and depend only on the surface type. In reality, it is likely that these parameters change locally, possibly depending on the pH of the snow, bromide to chloride ratio, bromide concentration, and sea ice thickness. The values could depend on season, and meteorology might also have some effect: For example for higher wind speeds, wind pumping (Colbeck, 1997) may replenish the releasable bromide more quickly. This may also be the explanation for the large modeled BrO VCDs unobserved by the satellite near Hudson Bay and Baffin Bay. The local values for the releasable bromide and replenishment timescale there might be smaller than the global values assumed in this model, perhaps due to the lower salinity of the sea water at Hudson Bay, or the local values could become smaller in course of the year. For instance, the larger bromide reservoir, the deeper layers of the snow or sea ice, from which the readily available surface bromide is replenished, could be depleted during the year at very active locations like Hudson Bay.

The simulations are not suitable to completely explain all ODEs at Eureka and Nord Station, even when MY ice and snow-covered land are assumed to have the same amount of releasable bromide as FY ice. This might be due to the stochastic character of the process or due to a missing mechanism in the model, perhaps blowing snow or an insufficient grid resolution, since the topography at these locations is very complex and not all sea ice near Eureka is resolved. Over Greenland, BrO is not observed by TROPOMI, but the models sometimes find enhanced BrO VCDs which are typically produced in Baffin Bay and are advected to Greenland. Large BrO VCDs at Baffin Bay occur more often in the model in comparison to the satellite, which could explain the unobserved BrO VCDs in Greenland, but meteorological inaccuracies cannot be ruled out.

Considering the success of the finite bromide simulations, the releasable bromide and its replenishment on FY ice appear to be reasonably chosen. However, the amount of bromide stored in the top centimeter of snow covering sea ice as measured by Pratt et al. (2013) is about 20 times higher than the value of releasable bromide on FY ice assumed in this work. Assuming that the value measured by Pratt et al. (2013) is representative for the wider Arctic, it is likely that there is a mechanism limiting the amount of releasable bromide such as the rise of the pH. The improvements of the agreement of the medium and high bromide simulations with observations near MY ice suggest that MY ice is a significant bromine source, perhaps comparable to FY ice. Releasable bromide on snow-covered land is likely to be smaller than that chosen for the medium bromide simulation, which assumed 50 % of the value on FY ice, since results at locations such as Utqiaġvik or Pallas are slightly worse in comparison to the low bromide simulation and overestimations of BrO VCDs near Baffin and Hudson Bay increase, especially over land. This is consistent with the amount of bromide stored in the top one cm of tundra surface snow, which is about the same value of releasable bromide on snow-covered land assumed in this work. Thus, it is likely that the bromide concentration in the tundra snow limits the amount of releasable bromide.

The bromine emissions model is currently agnostic to the exact mechanism of the releasable bromide and its replenishment, which could be modeled in the future. For example, the replenishment rate could be considered dependent on meteorological variables, for instance by estimating wind pumping, or an attempt to model the replenishment of bromide from open leads may be made.

It is worthwhile to mention that the new emission scheme presented in this work does not cause additional computational cost and it is easy to implement into any three-dimensional model. Thus, there is a high potential to use the new finite bromide model, which is superior to the infinite FY ice bromide assumption at almost all observation sites, in future studies.

*Code availability.* Both the model data and code are available upon request.

*Author contributions.* MH performed the simulations and model and software development. MS and TW provided the observations. CB performed the spectral analysis. MS, CB and SW conducted the data analysis. TW evaluated the meteorology. UP and EG conzeptualized and supervised the project. EG was responsible for funding. All authors contributed to writing and revising the paper.

*Competing interests.* The authors declare no competing interests.

*Acknowledgements.* Funding by the Deutsche Forschungsgemeinschaft (DFG, German Research Foundation) – Projektnummer 85276297 and through HGS Math-Comp is gratefully acknowledged. The authors appreciate support by the State of Baden-Württemberg through bwHPC and the DFG through grant INST 35/1134-1 FUGG, allowing the authors to conduct simulations using the bwForCluster MLS&WISO Production. In-situ ozone data for Utqiaġvik, Summit and Eureka were obtained from the NOAA/ESRL Global Monitoring Division. In-situ ozone data for Utqiaġvik, Summit and Eureka were obtained from the EBAS database. Vertical ozone sonde profiles are available from the 675 World Ozone and Ultraviolet Radiation Data Centre. ERA5 hourly single and pressure levels data were provided courtesy of ECWMF. CAM-Chem chemistry data were obtained from UCAR/ACOM. The authors acknowledge the use of the WRF-Chem preprocessor tools (mozbc, bio_emiss, anthro_emiss and mozartpreprocessor) provided by the Atmospheric Chemistry Observations and Modeling Lab (ACOM) of NCAR. The authors thank ESA and the S-5P/TROPOMI level 1 teams for providing the TROPOMI level 1 data. The authors thank Steffen Beirle for the development of the fitting routines used to produce the satellite BrO VCDs shown in this work.

## 680 Appendix A: Changes to the GOME-2 BrO retrieval

The retrieval of tropospheric BrO from satellite data is based on the work by Sihler et al. (2012), but several adjustments of their algorithms had to be made to adapt them to the much higher resolution of TROPOMI. These changes will be explained in the following.

### A1 Evaluation of TROPOMI spectra

Several small adjustments were made to the underlying DOAS BrO retrieval. The largest difference is the change in the reference spectrum. Whereas Sihler et al. (2012) evaluated an irradiance spectrum, our retrieval uses an earthshine spectrum.

This was done due to a large across-track variability even after applying the corrections discussed by Sihler et al. (2012) and is done in similar fashion for other TROPOMI BrO retrievals (Seo et al., 2019). As reference region for the calculation of an earthshine spectrum we use an equatorial reference [$20°$ S to $20°$ N] around the whole globe. For the BrO fit itself we updated most of the cross sections with more recent fit settings as shown in Tab. 2 while the fit window as well as the absorbers are identical.

## A2 Column separation

A major change to the study by Sihler et al. (2012) is the algorithm to separate the stratospheric and tropospheric part of the BrO column. Although the conceptual idea stayed the same, the implementation of the algorithm was changed quite drastically to be applicable to TROPOMI. The new algorithm can be divided into three parts:

1. A subset of measurements avoiding potential interferences with anthropogenic $NO_2$ is selected following the work by Sihler et al. (2012).

2. The measurements are partitioned into subsets of similar size in the hyperplane spanned by the solar zenith angle (SZA) and the $NO_2$ vertical column density. For this part of the algorithm major changes were necessary. We replaced the iterative shifts of the decision boundaries as described in the appendix of the study by Sihler et al. (2012) by a slightly modified version of an k-means++ (Arthur and Vassilvitskii, 2007) algorithm, where we emphasized the penalization of unequal cluster sizes as visualized in Fig. A1. This has several advantages over the old clustering procedure:

   - The runtime complexity was reduced from $\mathcal{O}(n^2)$ to $\mathcal{O}(n \log n)$. Considering the large increase in measurements each day, this reduction in computational complexity can be considered necessary. After applying the aforementioned selection criteria, the number of pixels to process each day is approximately 500 times larger than for GOME-2.

   - The amount of hyperparameters was reduced to only one, which is identical to the number of clusters. Whereas the old algorithm had to make additional empirical inferences, e.g., the incremental changes per iteration to partition the data, the new version of the clustering does not need any additional input besides the number of clusters.

   - The new clustering is less susceptible to hyperparameter changes, i.e., changes in the number of clusters. Whereas changes to the number of clusters could induce a variance in the retrieved tropospheric BrO of up to 10 % for the old algorithm (Sihler et al., 2012), the new version only introduces a variance of about 1-2 % for a sensible parameter range (10-500 clusters). This means that the number of clusters does not need to be tuned as precisely, although this in part is also due to the higher number of measurements from TROPOMI and therefore the better statistics per cluster.

   Moreover, the division of the measurements into viewing zenith angle (VZA) bins can be omitted with TROPOMI. While for GOME-2 a weak dependence on the VZA was detected, this was no longer the case for TROPOMI. This can probably be ascribed to the different equator crossing times of the two satellite instruments, where especially for

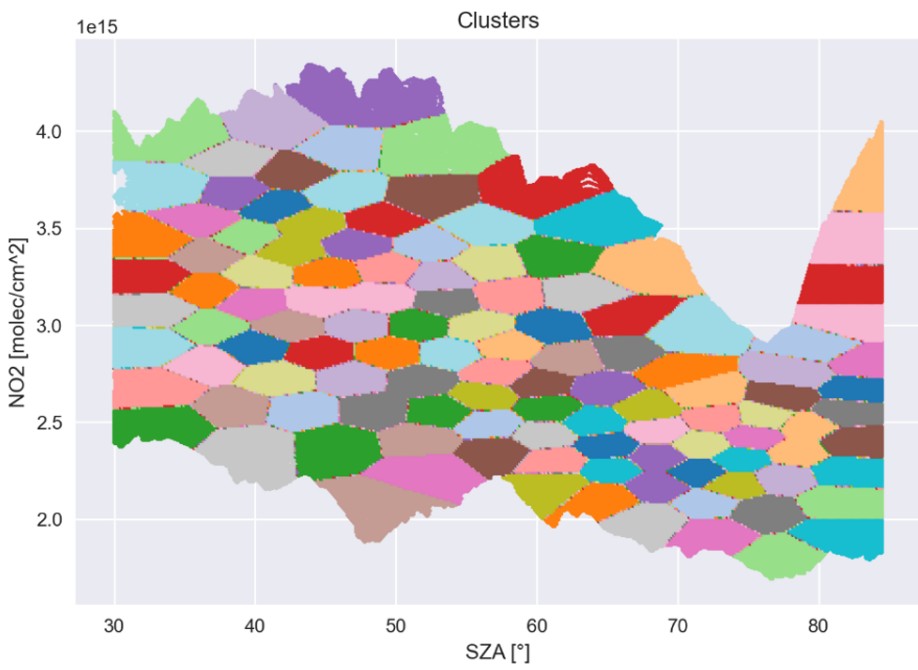

**Figure A1.** Visualization of decision boundaries used to sort measurements into clusters.

the GOME-2 instrument with an equator crossing time of 9:30 h local solar time, the edges of the swath could exhibit greater variance in the actinic flux.

3. The clustered measurements are then used to determine the stratospheric BrO background. The underlying assumption is that for enough measurements in each clusters, the stratospheric BrO signal would dominate. For a purely stratospheric signal, the BrO/O$_3$ ratio will then be dominated by the BrO uncertainty, exhibiting an approximate Gaussian distribution $\phi(x) = \frac{1}{\sqrt{2\pi}}e^{-\frac{1}{2}(\frac{x-\mu}{\sigma})^2}$. Tropospheric enhancements of BrO will skew this distribution towards the right, which can be written in the form of $f(x) = \phi(x)\gamma(\alpha x)$, where $\gamma$ describes the asymmetry and $\alpha$ the skewness. Determining the mean $\mu$ of the underlying Gaussian distribution $\phi(x)$ therefore allows the approximation of the stratospheric BrO signal in each cluster. Previously, this was done via an iterative cropping of distribution until a certain skewness threshold was achieved Sihler et al. (2012). The problem with this approach is the calculation of the distribution mean $\sigma$ via the arithmetic mean, which only holds if the distribution indeed is Gaussian. To reduce the systematic error from this assumption, also measurements on the left tail of the distribution are cropped. However, with TROPOMIs higher signal-to-noise ratio, the variance of the distribution is lower than for GOME-2 and this approach leads to a systematic overestimation of the mean $\mu$, especially for clusters with only a weak tropospheric enhancement. Therefore, for TROPOMI, we updated this part of the algorithm. Instead of using an iterative approach to filter out tropospheric signals, we now fit an asymmetric Gaussian based on the study by Beirle et al. (2017). The parameters of this Gaussian are empirically tuned, which re-

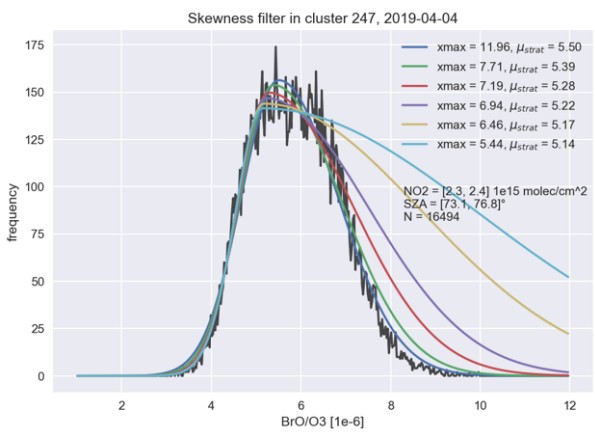 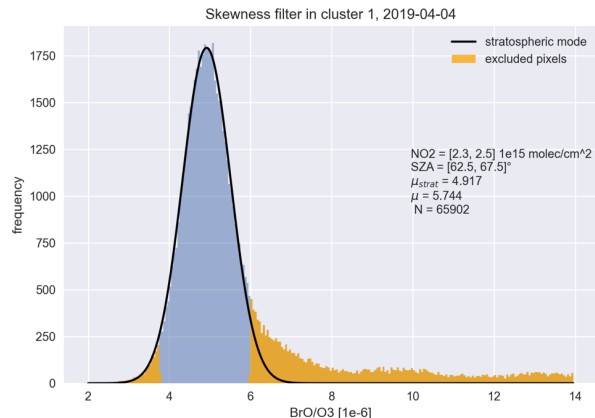

**Figure A2.** Different parameter choices for the stratospheric background calculation visualized (left) result of final parameter choice (right).

sults in an improved fit of the underlying Gaussian $\phi(x)$ shown in Fig. A2. In a last step, the stratospheric background calculated for each cluster are used to interpolate the stratospheric BrO background over the whole domain (Sihler et al., 2012).

## A3 Derivation of air-mass factors

The calculation of air-mass factors and the surface sensitivity filter is almost identical to the study by Sihler et al. (2012). We
just added more nodes to our radiative transfer LUT to account for the higher pixel density of TROPOMI, especially at the edges of the swath. The differences are summarized in Tab. A1.

**Table A1.** Summary of modeled geometries and additional geometry nodes used for TROPOMI

| Parameter | GOME-2 nodes | TROPOMI additional nodes |
|---|---|---|
| SZA [°] | 28, 44, 56, 64, 66, 68, 72, 76, 80, 82, 84, 86 | 10 |
| VZA [°] | 0, 16, 32, 48 | 56, 64, 68, 70 |
| RAA [°] | 0, 20, 32, 36, 44, 48, 52, 56, 60, 64, 116, 120, 124, 128, 132, 136, 144, 148, 160, 180 | 68, 72, 108, 112, 140 |
| Elevation [km] | 0, 1, 2, 3, 4, 5, 6 | - |

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
