# Peer review of "Ozone depletion events in the Arctic spring of 2019: A new modeling approach to bromine emissions"

_Atmospheric Chemistry and Physics, 2022_

## Author Response (AR2)

We thank the reviewers for their valuable comments which lead to a great improvement of the present submission. We revised the manuscript where modifications of the text are marked in blue color. In particular, we address the comments in detail as follows.

**Reviewer #1**

The paper details the use of 3D modelling (WRF-Chem) to reproduce Arctic bromine chemistry and associated impacts (ODEs). The source of the bromine is snow within sea ice regions and within 300 km of the coast and in contrast with other studies this is a limited, replenishable, source rather than an infinite one. The effectiveness of the model is evaluated using BrO measurements from TROPOMI, in-situ ozone measurements at various Arctic sites and ozone sonde data. Importantly, unlike many other 3D modelling papers, this paper does not consider a "blowing snow" source. The authors find the new parameterization improves agreement with observations. The extent of this improvement varies from site to site depending on the sea ice conditions prevalent at those sites. The work is an interesting contribution and merits publication, however I would like to see the authors motivate their chosen snowpack bromide values using the larger data sets available rather than the limited dataset of Pratt et al 2013.

Specific Comments:

The salinity assumption detailed in line 218 is odd. Pratt et al 2013 is not a comprehensive snow data set for the Arctic and has a very limited number of samples. Krnavek et al 2012 would be a much better resource as the sample size is much larger and more likely to be representative. Then the value from Pratt et al is discarded in favor of a value that seems to be chosen rather arbitrarily. Kranvek et al as well as Peterson et al 2019 also have values for bromide in different ice regions which would be useful to improve your values of initial releasable bromide. Assuming the values for MYI regions and land are related to the first year ice region value seems like another arbitrary choice, particularly when measurement data are available. The paper would be improved if the authors motivated their chosen values using existing more comprehensive snowpack bromide measurement datasets.

**Authors' Answer:**
Thank you for the suggestion. We added the following text to the salinity discussion starting on line 225:

"Using the above calculation, Krnavek et al. (2012) measured bromide column densities on FY ice of $1.2 \times 10^{14}$ molec/cm2 to $1.8 \times 10^{17}$ molec/cm^2, with a median of $1.2 \times 10^{15}$ molec/cm^2 on thick FY ice, so that the values on FY ice used in this work are consistent with the lower range of values found by Krnavek et al. (2012). Peterson et al. (2019) found lower halide concentrations due to measuring thicker sea ice (> 1 m), and they found an average bromide column density of $3.6 \times 10^{14}$ molec/cm^2 , which is consistent with the value used in this work."

**Reviewer #1:**
Line 166: Why were observations at other sites not used?

**Authors' Answer:**
We considered the other observation sites only at a later stage of our work, after the rescaling of the initial values was already done and the first simulations were conducted. Since the rescaling has very little effect after the first ODEs occur in early March, we did not think it was necessary to redo the simulations with a different rescaling considering all sites. No change were made in the manuscript.

**Reviewer #1:**
Line 268: This statement concerning upward migration is only true if the snowpack is sufficiently shallow. In deeper snowpacks, the surface salinity is likely not impacted by brine migration and thus the bromine source would be limited. Domine et al. (2004) suggest this point is about 17 cm.

**Authors' Answer:**
Thank you, we corrected the sentence as follows: "Upward migration of sea salt from sea ice is thus indeed a practically unlimited bromide source for modeling purposes, assuming the values found by Pratt et al. (2013) and the median and higher values measured by Krnavek et al. (2012). However, upward migration is likely only effective for sufficiently shallow snowpacks, which was suggested to be approximately 17 cm by Domine et al. (2004). So the assumption of an unlimited bromide reservoir could be unproblematic for shallow snowpacks, but is likely to be wrong for deeper snowpacks."

**Reviewer #1:**
Figures 9 and 12: It is hard to get what I am supposed to take away from these figures. Perhaps it might help to plot to difference between the modelled and satellite observations which would allow you to reduce the number of panels as well as allow the reader to more easily spot the areas of agreement/disagreement.

**Authors' Answer:**
If we plot the differences of satellite and model, it is best to show the absolute values of the satellite observations only as well, in order to be able to see where bromine events are occurring, which would not be obvious from showing the differences only. Because of that, the total amount of panels would remain the same.

Also, we do not think it is necessarily easier to spot areas where the model finds events and the satellite does not, since you would always have to look at the satellite only figure in order to confirm if the satellite found any events at that location. The opposite case, events found by the satellite but not by the model, would be very hard to see, since the reader has to subtract the difference between satellite and model from the satellite only figure in their head in order to determine if the model completely missed that bromine event.

For these reasons, we believe the current figures are appropriate and no changes in the manuscript were made.

**Reviewer#1:**

Technical Corrections

Line 240: Delete "the" between full and value

Authors' Answer:
We removed the word "the".

The following references for Reviewer #1 were added to the revision:

Krnavek, L., Simpson, W. R., Carlson, D., Domine, F., Douglas, T. A., & Sturm, M. (2012). The chemical composition of surface snow in the Arctic: Examining marine, terrestrial, and atmospheric influences. Atmospheric Environment, 50(0), 349–359. https://doi.org/10.1016/j.atmosenv.2011.11.033

Peterson, P. K., Hartwig, M., May, N. W., Schwartz, E., Rigor, I., Ermold, W., Steele, M., Morison, J. H., Nghiem, S. V, & Pratt, K. A. (2019). Snowpack measurements suggest role for multi-year sea ice regions in Arctic atmospheric bromine and chlorine chemistry. Elem Sci Anth, 7(1), 14. https://doi.org/10.1525/elementa.352

Domine, F., Sparapani, R., Ianniello, A., & Beine, H. J. (2004). The origin of sea salt in snow on Arctic sea ice and in coastal regions. Atmospheric Chemistry and Physics, 4(9/10), 2259–2271. https://doi.org/10.5194/acp-4-2259-2004

**Reviewer #2:**

In this manuscript, the authors expand on an earlier study on modelling Arctic spring ozone depletion. The main change to the previous study is the change in bromine source at the surface, which now is no longer just from first year ice where it was assumed to be released from an unlimited reservoir, but is separated by FYI, MYI and snow on land and connected to a limited reservoir. The model results are compared to surface ozone and sonde measurements as well as satellite BrO columns, and the results are discussed on a station by station basis, focusing on the effects of different source regions.

The manuscript is well written and fits the scope of ACP. The new approach to bromine emissions from the surface is interesting, and the results are promising as agreement with observations is improved significantly. The proposed method is computationally inexpensive and could be implemented in other models as well which makes this study relevant for the community. I therefore suggest publication after minor revisions following my suggestions below.

Detailed Comments

**Reviewer #2:**

1. In the manuscript, the emphasis is on the distinction between limited and unlimited bromide reservoirs. However, the scenario "unlimited" also differs in that it only takes FYI into account as a source while the other scenarios also include emissions from MYI and snow on land. In my opinion, this mixing of two aspects, the limitation of the reservoirs and the inclusion of additional source types is unfortunate as it complicates interpretation of the results. Adding an additional model run with a limited reservoir for FYI and no other emissions would help to better separate effects.

Authors' Answer:

It is true that a simulation without sources on MY ice and snow would make the separation of effects easier, but we think that the sources of the low bromide simulation are small enough to serve as a proxy for a simulation without additional sources, so that a further simulation which would take a significant effort is not necessary. No change was made in the manuscript.

**Reviewer #2:**

2. As also pointed out by the authors, a number of ad hoc decisions on values and free model parameters had to be made in the model, which probably cannot be avoided but carries the risk of optimisation of results towards the stations used for validation. This includes the values chosen for the reservoirs, the distance parametrisation for snow on land, the replenishment time and also the ozone scaling in the model.

Authors' Answer:
We conducted three simulations in order to optimize the ozone scaling (all using the infinite bromide assumption), but only data in Utqiagvik and Eureka were was used for this and as we found later, the scaling had very little effect after the ODEs started in early March, which is the time period of interest. We conducted exactly three simulations with the finite bromide assumption, all presented in the paper. There was no optimization for the values on FY ice and the replenishment time, the values used in the three finite bromide simulations was our very first guess. The initial bromide and distance parameterization was varied in the three finite bromide simulation, so that was an optimization effort motivated by the underestimated ozone depletion at Eureka. Thus, our first guess, the low bromide simulation, already works very well. No change was made in the manuscript.

**Reviewer #2:**

3. The exclusion of blowing snow as a source of bromine is a limitation but in view of the good agreement between model and measurements could be seen as indication for a limited importance of blowing snow for bromine release. However, this could also be linked to the choice of validation stations.

Authors' Answer:
We used data from all observations available to us in and near the Arctic region, so we do not think the good agreement is due to our choice of validation stations. The limited importance of blowing snow was also found by Marelle et al. (2021), so we see this work as a confirmation of their findings. No change was made in the manuscript.

**Reviewer #2:**

4. The comparison to satellite data is less clear than the one for the ozone observations. I would suggest adding a table with correlations and differences, similar as for the other comparisons.

Authors' Answer:
We added Tab. 5 to show the correlations of model and satellite BrO VCDs at the measurement sites for the ozone mixing ratio. A short discussion of the findings was also added, beginning on line 528.
In order to calculate these, we interpolated model and satellite BrO VCDs to the same grid, then interpolated model values in time to satellite time points and calculated the correlation in time for every grid point. Finally, we took the nearest neighbour to the measurement sites in space to receive the correlation at that location.

**Reviewer #2:**

5. The description of the new TROPOMI BrO product is very brief and vague – is that the Sihler et al/ algorithm applied to TROPOMI data, or was the algorithm and the thresholds used changed? If more changes were applied to the Sihler et al. retrieval, then this should be documented, for example in an appendix.

Authors' Answer:
Indeed, there were more changes applied to make the algorithm from Sihler et al. (2012) work for TROPOMI data. We decided not to go into too much detail here since the focus of the paper is the simulation results of the model. We added an appendix A describing the changes to the algorithm compared to Sihler et al. (2012) just before the references.

**Reviewer #2:**

6. Why was an O4 cross-section at the unrealistically low temperature of 203 K used?

Authors' Answer:
The low temperature cross section was chosen due to consistency reasons with the BrO fit for volcanic plumes deployed in Warnach (2022). There such low temperatures are to be expected, especially for larger volcanic eruptions.
Nevertheless, this temperature region works for the arctic BrO, too. There the vertical temperature average weighted by the O4 column is only about 30K higher during the relevant part of the year. Additionally, the difference between the cross sections between the 203K and the 293K is very marginal in the chosen wavelength interval.

**Reviewer #2:**

7. Are the two Pukite terms for ozone?

Authors' Answer:
Yes. We added this information to the paper in Table 2 together with a reference (Pukite et al. 2010)

**Reviewer #2:**

8. Is the uncertainty discussion given for individual TROPOMI pixels?

Authors' Answer:
Correct, the discussion concerns the uncertainty of individual TROPOMI pixels.

**Reviewer #2:**

9. Line 328 – I would call this a small rate of false positives but maybe this is a matter of definition

Authors' Answer:
Thanks for pointing this out. Indeed, since the purpose of this step of the algorithm is to filter pixel not sensitive to the ground, this should be called a small rate of false positives. We adjusted this in the paper.

**Reviewer #2:**

10. In tables 3 and 4, I would suggest to use bold face to highlight the best values for each comparison instead of a certain model setup which can readily be identified by its position in the table.

Authors' Answer:
We highlighted the best values as suggested by the reviewer.

11. line 593 – not sure what the authors are trying to say with the "chaotic component" – I assume that in the model, the initialisation of the bromine explosion is deterministic and not driven by random processes?

Authors' Answer:
We mean chaotic in the sense that small changes, e.g., in the initial conditions, may lead to large changes in the occurrence of ODEs. The model is still deterministic in any case. We extended line 593 for clarification: "It should be noted that the bromine explosion as an autocatalytic process has an inherently chaotic component, i.e., for example, small changes in the initial conditions may lead to large differences in the occurrence of ODEs."

References used by the authors in this authors' answer

Pukite, J., Kühl, S., Deutschmann, T., Platt, U., and Wagner, T.: Extending differential optical absorption spectroscopy for limb measurements in the UV, Atmospheric Measurement Techniques, 3, 631–653, https://doi.org/10.5194/amt-3-631-2010, 2010.

Marelle, L., Thomas, J. L., Ahmed, S., Tuite, K., Stutz, J., Dommergue, A., et al. (2021). Implementation and impacts of surface and blowing snow sources of Arctic bromine activation within WRF-Chem 4.1.1. *Journal of Advances in Modeling Earth Systems*, 13, e2020MS002391. https://doi.org/10.1029/2020MS002391

*Warnach, S., 2022. Bromine monoxide in volcanic plumes: a global survey of volcanic plume composition and chemistry derived from Sentinel-5 Precursor/TROPOMI data. Heidelberg: Universitätsbibliothek Heidelberg.*

Note that the reference Warnach (2022) is only used in this rebuttal and the paper Marelle et al. (2021) is already cited in the original submission. Pukite et al. (2010) has been added to the revised paper.